# Intercomparison of middle-atmospheric wind in observations and models

Rolf Rüfenacht[1,*], Gerd Baumgarten[1], Jens Hildebrand[1], Franziska Schranz[2], Vivien Matthias[1], Gunter Stober[1], Franz-Josef Lübken[1], and Niklaus Kämpfer[2]

[1]Leibniz-Institute of Atmospheric Physics at the Rostock University, Kühlungsborn, Germany
[2]Institute of Applied Physics, University of Bern, Bern, Switzerland
[*]now at: Federal Office of Meteorology and Climatology MeteoSwiss, Payerne, Switzerland and Institute of Applied Physics, University of Bern, Bern, Switzerland

*Correspondence to:* rolf.ruefenacht@iap.unibe.ch

**Abstract.** Wind profile information throughout the entire upper stratosphere and lower mesosphere (USLM) is important for the understanding of atmospheric dynamics but became available only recently, thanks to developments in remote sensing techniques and modelling approaches. However, as wind measurements from these altitudes are rare, such products have generally not yet been validated with (other) observations. This paper presents the first long-term intercomparison of wind observations in the USLM by co-located microwave radiometer and lidar instruments at Andenes, Norway (69.3° N, 16.0° E). Good correspondence has been found at all altitudes for both horizontal wind components for nighttime as well as daylight conditions. Biases are mostly within the random errors and do not exceed 5-10 m/s which is less than 10% of the typically encountered wind speeds. Moreover, comparisons of the observations with the major re-analyses and models covering this altitude range are shown, especially also with the recently released ERA5, ECMWF's first re-analysis to cover the whole USLM region. The agreement between models and observations is very good in general, but temporally limited occurrences of pronounced discrepancies (up to 40 m/s) exist. In the article's appendix the possibility of obtaining nighttime wind information about the mesopause region by means of microwave radiometry is investigated.

## 1 Introduction

Measurements of the wind field in the upper stratosphere and lower mesosphere (USLM) are challenging. The consequence is a substantial data gap between 10 and 0.03 hPa (∼32 to 70 km). Figure 1 summarises the altitude coverage of the currently existing wind measurement techniques. The widely used radar techniques can usually not assess the USLM due to the lack of backscatterers (charged particles, turbulent structures at scales of the radar wavelength). Only in the event of strong particle precipitations have measurements down to 60 km been reported (e.g. Nicolls et al., 2010; Shibuya et al., 2017, for an encompassing overview on radar observation techniques refer to e.g. Hocking (2016)). On the other hand the transport of in situ sensors to these altitudes cannot be achieved with radiosoundings or airplanes. For many years rocket-aided measurements were thus the only way to overcome this data gap. Such observations have been carried out since the 1960s (National Research

Council, 1966). They offer high vertical resolution but are very manpower- and cost-intensive. Hence, measurements are only made on campaign basis so that the data set consists of snapshots of the atmosphere highly scattered in time.

In the last decade two new techniques that achieve wind observations throughout the entire USLM, independent of daylight conditions, became operational: Doppler microwave wind radiometry and middle-atmospheric lidar spectroscopy (Rüfenacht

et al., 2012; Baumgarten, 2010). While wind radiometry was developed from scratch, the lidar technique could benefit from earlier works on nighttime stratospheric wind measurements by lidar (Chanin et al., 1989; Souprayen et al., 1999; Tepley, 1994; Friedman et al., 1997). Due to the novelty of the two approaches and the absence of satellite data, thorough validations with two independent measurement techniques synchronously assessing the same atmospheric volume are at a very early stage. Such intercomparisons at multi-instrument sites are a key activity of the Horizon 2020 project ARISE[1] (Blanc et al., 2017). Previously,

Lübken et al. (2016) presented comparisons between observations with the ALOMAR RMR lidar and eight nighttime starute[2] soundings by rockets. In most cases good correspondence between both techniques has been found, even in the small-scale structures. However, for some soundings the intercomparisons suffered from differing spatial sampling. Such sampling issues are closely related to the snapshot nature of rocket measurements. They can largely be overcome when comparing techniques which both allow longer observation times, as it is the case for the radiometry and lidar approaches, because atmospheric

inhomogeneities tend to average out over time. The RMR wind lidar and the ground-based wind radiometer WIRA have been operated next to each other at ALOMAR observatory in Andenes, Norway (69.3° N, 16.0° E) for a 11 months intercomparison period between 1 August 2016 and 30 June 2017. During this period, 518 hours of coincident measurements of sufficient duration[3] and an uninterrupted series of 187 hours of continuous day and night observations have been recorded.

In parallel to the development of new measurement techniques most important general circulation models for numerical

weather prediction and re-analysis have extended their lids further into the middle atmosphere owing to the broad evidence for the influence of middle-atmospheric dynamics on tropospheric weather and climate (e.g. Baldwin and Dunkerton, 2001; Scaife et al., 2008; Kidston et al., 2015; Garfinkel et al., 2017). The most recent example of this trend towards increased model tops is ECMWF's ERA5 re-analysis released in July 2017, which extends up to 0.1 hPa, one pressure decade higher than its predecessor ERA-Interim. In addition to the improvement of the tropospheric forecast skills of the weather predictions, the

higher model lids made available re-analysis data for the stratosphere and mesosphere which are widely used in the research community. However, up to now, only few comparisons between wind observations and models exist[4]: Kishore Kumar et al. (2015) analysed the correspondence of fortnightly rocket wind soundings with the MERRA re-analysis and Hildebrand et al. (2012, 2017) showed comparisons between January nighttime lidar measurements and ECMWF operational analysis and forecast data. The present study will show for the first long-term intercomparisons between wind observations and state-of-the-art

models and re-analyses (ERA5, ECMWF forecasts, MERRA2, SD-WACCM) by using the 11 months quasi-continuous data set recorded by the microwave radiometer WIRA.

---

[1] http://arise-project.eu

[2] STAble Retardation parachUTE, i.e. an extra-stable falling target deployed by the rocket and tracked by ground-based radars (e.g. Schmidlin et al., 1985)

[3] Only measurements longer than 5 hours are considered in this study in order to mitigate effects of the different pointing of the instruments (see Sect. 4) and to guarantee stable radiometer retrievals.

[4] Le Pichon et al. (2015) noted that also middle-atmospheric temperature is a little-validated product.

In this paper the first part describes the measurement techniques for wind observations in the USLM (Sect. 2), the models and re-analyses used in this study (Sect. 3) and considerations to the effects of spatial and temporal sampling (Sect. 4). In Sect. 5 the intercomparisons between the coincident lidar and radiometer observations are presented alongside with short-term comparisons to models. Model validations on longer time scales are described in Sect. 6 before we draw the conclusions of our research in Sect. 7.

## 2 Measurement techniques

Fig. 1 illustrates the unique situation of Andenes, Norway (69.3° N, 16.0° E) where all currently available wind measurement techniques covering the gap region in the USLM are concentrated. Wind radiometer, lidar and meteor radar are all contributing to the before-mentioned ARISE project (Blanc et al., 2017).

### 2.1 Doppler microwave radiometry

Microwave radiation is emitted in transitions between rotational quantum levels of molecules with electric (or magnetic) dipole moment, which are present in the gap region for wind observations in the USLM. Ground-based microwave radiometers have been widely used to determine the concentrations of the emitting molecules, e.g. water vapour, ozone or carbon monoxide (Lobsiger, 1987; Nedoluha et al., 1995; Forkman et al., 2003; Palm et al., 2010; Fernandez et al., 2016). Technical developments, especially in the field of high-frequency low-noise amplifiers and spectrometer stability/resolution, now enable the determination of the wind-induced Doppler shift of these emission signals. Altitude-dependent information is retrieved thanks to the pressure broadened nature of the emission spectra.

First measurements of profiles of horizontal wind by ground-based microwave radiometry had been presented by Rüfenacht et al. (2012). Recently, another similar instrument became operational and is currently set up at Maïdo observatory on La Réunion island (Hagen, 2015). The possibility for spaceborne wind observations with a comparable approach has also been demonstrated: SMILES was operated during 7 months onboard the International Space Station (ISS) (Baron et al., 2013) and an early-stage project for a satellite microwave limb sounding wind instrument exists (Baron et al., 2015).

The ground-based Doppler microwave wind radiometer WIRA, which contributes to the present study, is an upgraded version of the instrument presented in Rüfenacht et al. (2012). For the determination of wind profiles the Doppler shifts of signals from opposite azimuths in 68° off-zenith direction are compared. The retrieval algorithm which is based on the optimal estimation method (Rodgers, 2000) is similar to the version in Rüfenacht et al. (2014) with a constant zero wind a priori profile. In this study we use identical a priori standard deviations for both horizontal wind components equivalent to 4 times the standard deviation of 6 years of zonal wind data from ECMWF. In order to account for the high nighttime ozone concentrations which occur in the mesopause region the retrieval algorithm has been improved according to Rüfenacht and Kämpfer (2017). It now uses distinct ozone a priori profiles for nighttime or sunlit periods at the mesopause, discriminated by the sunrise/sunset at 100 km altitude. Based on considerations to atmospheric physics/chemistry and radiative transfer as well as on the comparisons of the day/night differences in the radiometer observations and models presented in Rüfenacht and Kämpfer (2017) the authors

judge now also the nighttime observations of mesospheric wind by WIRA to be largely bias-free. This quality is intended to be confirmed by the first instrumental intercomparisons carried out in the present study.

The wind radiometer WIRA can provide zonal and meridional wind profiles with a vertical resolution of 10-16 km with minimal integration times of around 5 hours. The trustworthy altitude range, i.e. where the measurement response is $> 0.8$, the altitude resolution is $< 20$ km and the altitude accuracy is $< 4$ km (for details see Rüfenacht et al., 2014), typically extends from 7 to 0.03 hPa ($\sim 35$ to 70 km). Microwave radiometers can be highly automatised and have the ability to pursue the measurements during overcast conditions what leads to near-continuous time series of observations, a characteristic that will be exploited for the model validations during almost a full annual cycle presented in Sect. 6.

## 2.2 Middle-atmospheric wind lidar

Lidar systems with powerful lasers, large telescopes and sensitive detection optics are able to get usable molecular backscatter from altitudes up to 70 to 80 km. When high frequency stability and calibration standards are met, wind can be determined from the Doppler shift of the backscattered signal. This is achieved by relating the recordings of a channel containing elements with frequency-dependent transmission in its receiver optics with recordings of a reference channel without such elements.

The first observations of zonal and meridional wind by lidars covering the entire gap region in the USLM have been presented by Baumgarten (2010) using the Rayleigh/Mie/Raman (RMR) lidar at ALOMAR, the instrument which is contributing to the present study. Validations of the method for nighttime observations have been presented by Hildebrand et al. (2012) and Lübken et al. (2016). Measurements of temperatures and wind during day and night are presented in Baumgarten et al. (2015). Recently, middle-atmospheric wind measurements from a similar instrument have been reported by Yan et al. (2017).

The ALOMAR RMR-lidar obtains wind information by single-edge iodine absorption spectroscopy on the 532 nm signal from 20° off-zenith. Injection-seeding of the transmitting lasers by a highly frequency-stable continuous wave laser, real-time monitoring of the wavelength transmitted to the atmosphere as well as regular calibrations of the frequency dependence of the transmission through the receiver optics and the iodine vapour cell assure the accuracy of the wind observations. Moreover, the exactitude of the calibration is optimised by performing 1 hour of vertical wind observations at the beginning of each measurement run (for details see Baumgarten, 2010; Hildebrand et al., 2012).

Thanks to its narrow field of view (100 $\mu$rad) and the possibility to add a Fabry-Pérot etalon with high finesse to the optical path the system has full daylight capability, which is especially valuable at high latitudes during polar day conditions. The daylight mode is automatically enabled when the solar elevation is higher than 4° below the horizon, (i.e. sunrise at 16 km). The frequency-dependent optical properties of the etalon and its effect on the different beam paths need to be accurately known from calibration measurements, what increases the complexity of the daylight wind measurements. In the previous rocket-lidar intercomparisons by Lübken et al. (2016) only nighttime wind profiles have been investigated. Daylight profiles will for the first time be validated in the present study.

The ALOMAR RMR lidar delivers wind profiles with very high vertical and temporal resolution (natively 150 m and 5 minutes). Some binning of the data is usually applied for increasing the signal-to-noise ratio. The measurement uncertainty depends on altitude and ranges from about 1 to 10 m/s at altitudes between 50 and 80 km for integration times of 1 hour

and vertical resolutions of 2 km. As for any middle-atmospheric lidar, the operation of the RMR lidar is limited to clear-sky conditions.

## 2.3 Meteor radar

Measurements from the Andenes meteor radar will be used in the present study although it is not directly covering the gap region for wind observations in the USLM. Indeed, the lowest altitudes covered by the meteor radar are adjacent to the upper-most levels covered by WIRA and the RMR lidar, at least in good observation conditions (reasonably low tropospheric water content for the radiometer, no cirrus clouds for the lidar). Hence, the meteor radar can help in the interpretation of the wind data at the highest levels of the USLM gap region.

Meteor radars providing wind observations through the evaluation of the drift of meteor trails in the wind field of the mesosphere/lower thermosphere region are a well-established technique (Hocking et al., 2001a; Jacobi, 2011; Fritts et al., 2012; Iimura et al., 2015). The reliability has also been demonstrated in recent comparisons to other remote sensing techniques and the Navy Global Environmental Model (NAVGEM) (Reid, 2015; McCormack et al., 2017; Wilhelm et al., 2017).

The Andenes meteor radar operates at 32.55 MHz with a peak power of 30 kW. The typical altitude range extends from 75 to 105 km. Details of the retrieval algorithm have been presented in Stober et al. (2017).

## 3 General circulation models, re-analyses and geostrophic wind

### 3.1 ECMWF ERA5 and forecast data

The recently released ERA5 re-analysis (Hersbach and Dee, 2016) is the European Centre for Medium-Range Weather Fore-cast's (ECMWF) first re-analysis to extend throughout the mesosphere up to 0.1 hPa. It provides hourly output so that a good temporal match with the observation data can be achieved. At the time of writing only the data prior to 31 December 2016 had been released. Therefore, alongside the ERA5 high-resolution (HRES) data, hourly forecasts (FC) from IFS cycles 41r2 (from 1 Aug 2016 to 21 Nov 2016), i.e. the same cycle as for the ERA5 re-analysis, and 43r1 (from 22 Nov 2016 to 30 Jun 2017) are used in the present study (ECMWF, 2017). ECMWF models and re-analyses use a 4DVAR assimilation scheme. The only source of upper air assimilations are infrared nadir sounders (AIRS, HIRS, IASI)(Dragani and McNally, 2013).

### 3.2 MERRA2

The Modern-Era Retrospective analysis for Research and Applications-2 (MERRA2) is the current re-analysis of NASA's Goddard Earth Observing System-5 (GEOS-5) model with 3-hourly output extending up to 0.01 hPa (Molod et al., 2015). It is the successor of the discontinued MERRA re-analysis used in the intercomparison study by Kishore Kumar et al. (2015). In contrast to ECMWF's models and re-analyses, MERRA2 also assimilates USLM observations from the microwave limb sounder (MLS) on the Aura satellite (Waters et al., 2006) in a 3DVAR assimilation scheme. A detailed description of the MERRA2 re-analysis was recently presented by Gelaro et al. (2017).

## 3.3 SD-WACCM

Thanks to its model top as high as $6 \cdot 10^{-6}$ hPa, the Whole Atmosphere Community Climate Model WACCM (Marsh et al., 2013) is a well-established data source for studies of middle-atmospheric dynamics. WACCM has a specified dynamics version named SD-WACCM (Lamarque et al., 2012; Kunz et al., 2011) which is suitable for intercomparisons with measurements. To constrain the dynamics of the model its temperature, horizontal wind and surface pressure fields are nudged with meteorological analysis data at every internal time-step. For the present study SD-WACCM was nudged to the GEOS-5 meteorological analysis data at every 30 minutes time-step. The nudging coefficient is 10% which means that the nudged fields are defined as a linear combination of 90% from the model and 10% from GEOS5 data (Brakebusch et al., 2013). This nudging is performed up to 50 km, then it linearly decreases in strength to zero nudging above 60 km. In the model gravity waves are parameterised, whereas planetary waves are resolved (Richter et al., 2010). SD-WACCM has a free running chemistry in the whole atmosphere.

## 3.4 Geostrophic wind from the MLS geopotential height field

The geostrophic wind describes the balance between Coriolis force and pressure gradient force and flows parallel to the isobars. This balance seldom holds exactly in the real atmosphere due to non-conservative forces. However, the geostrophic wind is a good approximation outside the tropics and therefore can be used for comparison with the other wind data in this study. Additionally, in contrast to the models previously described in this section whose assimilation schemes are primarily based on input from lower altitudes, this approach has a very direct connection between the wind field and observed data at the altitudes of interest.

Here, we calculate the geostrophic zonal ($u_g$) and meridional ($v_g$) wind from geopotential height (GPH) profiles of the Microwave Limb Sounder (MLS) on board the Aura Satellite (Waters et al., 2006; Livesey et al., 2015) by

$$u_g = -\frac{1}{f}\frac{\partial \Phi}{\partial y} \qquad v_g = \frac{1}{f}\frac{\partial \Phi}{\partial x} \tag{1}$$

where $\Phi$ is the geopotential, $f$ is the Coriolis parameter, and $x$ and $y$ are used to denote the partial derivatives $(a\cos\phi)^{-1}\frac{\partial}{\partial \lambda}$ and $a^{-1}\frac{1}{\partial \phi}$ where $\lambda$ is longitude, $\phi$ is latitude and $a$ is Earth's radius. Note that in this formulation friction, vertical advection and time tendency is neglected and that the geostrophic balance is assumed, i.e. the exact balance between Coriolis force and pressure gradient force. Therefore the geostrophic wind is directed parallel to the isobars and does not depend on curvature at all meaning that the air does not flow from high to low pressure. However, outside the tropics geostrophic wind can often be regarded as a reasonable approximation of the real wind.

MLS has a global coverage from 82°S to 82°N on each orbit and a usable height range from 261 to 0.001 hPa (11 to 97 km), with a vertical resolution of ~4 km in the stratosphere and ~14 km at the mesopause. Daily averages of version 4 MLS data were used and the most recent recommended quality screening procedures of Livesey et al. (2015) have been applied. The GPH observations have a precision between ±30 m at the tropopause and ±110 m at the mesopause level and a bias of 50 to 150 m in the troposphere and stratosphere and up to -450 m at 0.001 hPa (Froidevaux et al., 2006; Schwartz et al., 2008).

For the geostrophic wind estimation the original orbital MLS data are accumulated in grid boxes with $20°$ grid spacing in longitude and $5°$ in latitude and averaged over one day. This global smoothed data is then used to calculate the global geostrophic wind using equation 1. For the comparison with the local measurements the average geostrophic wind in the area $67° - 72°$N and $0° - 30°$E is chosen from the global calculations.

Some marginal aliasing effects on MLS data from the migrating tides can not be excluded. However, since Aura is in a sun synchronous orbit, its samples are stationary with respect to migrating tides. These should appear as constant offsets to the measurements at a particular latitude. Especially the effect of the diurnal tide which appears to be the strongest tidal component in the middle atmosphere is strongly reduced by the averaging over the measurements during the satellite's overpasses in the ascending and descending orbit spaced by 12 hours. A more detailed discussion on the impact of tides on MLS measurements can be found for example in Lieberman et al. (2015) and Xu et al. (2009). It should also be remembered that, in contrast to the mesopause region, tides are usually weak in the stratosphere and lower mesosphere (e.g. Baumgarten et al., 2018; Kopp et al., 2015; Sakazaki et al., 2018).

## 4 Spatial and temporal sampling of observations and models

A crucial aspect of intercomparisons of atmospheric data is to account for the different temporal and spatial sampling of the used instruments and models. Radiometer winds are retrieved on pressure levels while the lidar operates on a geometrical height grid, therefore all data are transformed to pressure coordinates according to the CIRA86 climatology (Fleming et al., 1990) for the respective day.

### 4.1 Horizontal and temporal sampling

The lidar observations are almost true point measurements, no horizontal averaging is involved. Due to their off-zenith nature the measured wind profiles are not completely vertical, e.g. the return signal at 70 km altitude originates from a point with a horizontal distance of 25 km to the observatory. Such small horizontal distances can safely be neglected in the present context. In contrast, the wind speeds obtained by the microwave radiometer involve measurements at two points horizontally distant by $2z \cdot \tan(68°)$, where $z$ is altitude above the observatory, i.e. 150, 250 and 350 km at altitudes of 30, 50, 70 km, respectively. The Andenes meteor radar obtains most meteor echoes from zenith angles between $50°$ and $60°$ leading to an average observation volume extent of about 160 km at an altitude of 70 km. Models and re-analyses also feature substantial horizontal smoothing so that particularly localised features are not captured. The data set with the lowest horizontal and temporal resolution are the geostrophic winds calculated from MLS geopotential heights (see Sect. 3.4).

In comparisons involving snapshot measurements such as rocket soundings or short-term lidar observations, effects from the previously described differing horizontal sampling can under no circumstances be neglected. Here, we consider only observations with integration times of more than 5 hours what mitigates the effects of different horizontal sampling. Over such long time periods, the average state of the middle atmosphere can normally be expected to be reasonably homogeneous over a few hundreds of kilometres of horizontal distance, especially in zonal direction. That this also holds for the meridional di-

rection is illustrated by the Supplement's Fig. S6 which shows that the effect of horizontal sampling by the wind radiometer induces biases in the order of not more than a few cm/s, even in the case of unusually strong meridional wind gradients over the observatory.

## 4.2 Vertical sampling

In contrast to horizontal smoothing the vertical structures are far more persistent in time. Therefore it is important to consider the limited vertical resolution of microwave radiometers which is in the order of 10-16 km for WIRA's wind observations. To compare high-resolution data $x_i$ (where $i$ stands for lidar or model data) to the observations from WIRA, these should be convolved with the radiometer's averaging kernels $\mathbf{A}$ (for details see Rüfenacht and Kämpfer, 2017) according to

$$x_{i,c} = \mathbf{A}(x_i - x_a) + x_a \tag{2}$$

where the a priori wind profile $x_a$ used by the radiometer is constantly zero. In the case of perfect instruments and models, all profiles $x_{i,c}$ and the observations by WIRA would agree within their random errors.

## 5 Intercomparisons for lidar operation times

In the sake of conciseness we mainly present averages over several days of measurements in this section. For the interested reader the wind profiles from the radiometer, the lidar and the models for each individual day and night of observation are re-printed in the Supplement's Figs. S1 to S4.

The longest uninterrupted lidar measurement of the intercomparison campaign took place from 3 to 11 February 2017 and lasted for 187 hours. The time series of the lidar and radiometer zonal and meridional wind profiles are shown in Figs. 2 and 3, respectively. These observations cover a particularly dynamic time period in the vicinity of a minor sudden stratospheric warming. For instance the zonal wind reverts its direction from -40 to 40 m/s within a few days. Obviously the lidar time series feature structures of atmospheric waves which can not be resolved by the microwave radiometer. Beyond this difference of resolution the time series from the two instruments correspond very well.

In Fig. 2 the strong westward winds on 3 February, the pronounced decrease in the wind velocities at stratopause level in the evening of 4 Feb along with the wind direction being inverted to eastward above 0.3 and below 3 hPa are all captured by both instruments. The same is true for the following westward acceleration on 5 and 6 Feb as well as the inversion of wind direction to eastward with two distinct wind speed maxima in the night of 8/9 and 10/11 Feb. It should be noted that also the altitudes and the timing of the features correspond very well. The most notable difference is that the radiometer wind at stratopause level on 4 Feb stays slightly negative whereas the lidar reaches a zero zonal wind situation what can mostly be attributed to the temporal averaging by WIRA which includes negative winds from before and after this event.

Similar considerations apply to the meridional wind in Fig. 3. Weak northward winds at the beginning of the time series are followed by a strong increase in wind speed between 4 and 7 Feb in both the radiometer and the lidar observations. The subsequent decrease and reversal to southward direction in the stratosphere during 8 Feb, another increase in northward wind

and finally the reversal to substantial southward winds below 0.3 hPa in the night from 10 to 11 Feb are also recorded by both independent instruments.

For more quantitative analyses the average daylight and nighttime wind profiles of this measurement period are presented in Fig. 4 and 5, respectively. The wind at each level has been averaged with the weights being the integration times $\Delta t_i$ of data sampled at this level during measurement $i$, i.e. $u_{\mathrm{avg}} = \sum(\Delta t_i \cdot u_i) / \sum \Delta t_i$. The errors of these average winds were accordingly calculated using Gaussian error propagation: $\sigma_{\mathrm{avg}} = \sqrt{\sum(\Delta t_i \cdot \sigma_i)^2} / \sum \Delta t_i$.

Generally daylight observations are more challenging for the lidar because of potential uncertainties related to the positioning of the additional Fabry-Pérot Etalon, whereas the nighttime radiometer wind measurements were known to be biased before the upgrade of the retrieval by Rüfenacht and Kämpfer (2017). Only time periods where lidar and radiometer are both in their respective daylight or nighttime configurations have been considered in the following analyses in order to independently validate both modes of each instrument.

For the daylight periods (Fig. 4) both observations and all models are in very close agreement at all altitudes and for both wind components. They all lie well within the random errors which are more than one order of magnitude smaller than the fluctuations of the atmospheric wind during this time. Only the meridional component of the geostrophic wind calculations shows an offset. This is, however, not surprising when considering the coarse resolution of this data set in combination with a very dynamic period with pronounced wind gradients around Andenes.

During nighttime (Fig. 5) up to 0.3 hPa all data sources agree within WIRA's observation errors or are very close. Above, substantial differences among the various data sources are visible. Notably the radiometer and lidar zonal winds agree throughout the entire altitude range and the ECMWF forecasts are close to the observations. In contrast, MERRA2 is slightly offset by 7 m/s and SD-WACCM by more than 15 m/s. For the meridional component, lidar and radiometer span the spread of offsets above 0.3 hPa which can reach up to 20 m/s. The model winds are scattered in between with ECMWF closest to the lidar, SD-WACCM closest to the radiometer and MERRA2 equally differing from both observed wind profiles. The lowermost meteor radar observations tend to indicate a preference for the low wind speeds measured by the wind radiometer. The reason for the disagreement between the meridional winds measured by the radiometer and the lidar at high altitudes could not be definitely identified. Although the radiometer and the meteor radar cover an observation volume of significantly larger extent than the lidar the discrepancy can most probably not be attributed to the different spatial sampling (see Supplement's Fig. S6). Nevertheless the substantial spread among the models and re-analyses indicate a rather heterogeneous atmosphere. A differing temporal evolution of the sensitivities of the lidar and the radiometer to these high altitudes might explain the dissent. Such effects could be introduced by temporally evolving cirrus or polar stratospheric clouds altering the transmission of the lidar signal or by variations of the mesospheric ozone concentration modulating the strength of the microwave emissions. In any case, as the zonal wind measurements are in very close agreement it is not believed that the high-altitude differences in meridional wind observations during these times indicate a fundamental instrumental problem.

The mean differences to the wind radiometer profiles over all measurements made during the entire 11 months intercomparison campaign are shown in Figs. 6 and 7. In total 348/326 hours of coincident daylight and 169/158 hours of nighttime zonal/meridional wind observations meeting the $> 5$ hour criterion have been made by the lidar and the radiometer. The slightly

reduced measurement time for the meridional component is due to the down-time of one power laser caused by its flash lamps reaching their end of lifecycle in October 2016. For an overview of the temporal distribution of lidar observations along with the dynamical situation around these recordings the interested reader is referred to Figs. 9 and 10 in Sect. 6 where days with lidar measurements are marked by green dots.

The daylight zonal winds of all models agree within the radiometer's random errors (Fig. 6). The lidar and the radiometer are in close correspondence up to 0.3 hPa, above the lidar has recorded more eastward winds with offsets of up to 10 m/s also slightly disagreeing with the models. One should however note the increasing uncertainty at these altitudes so that the error bands of WIRA and the RMR lidar almost overlap over the entire sensitive altitude range. The accordance for the daylight meridional component of the different data sources is very good at all altitudes.

A similar picture with again very close correspondence among the comparison data at all altitudes manifests for the nighttime zonal winds (Fig. 7). In contrast, the additional 54 hours contributing to this plot can obviously not eliminate the previously discussed meridional wind biases from the 104 hours of the long February observation shown in Fig. 5.

Despite the generally very good long-term agreement it should be noted that on shorter time scales the measurements may disagree with the models (see also Supplement's Figs. S1 to S4). A particularly illustrative example of this situation is

presented in Fig. 8. It shows the zonal wind profiles for the night from 4 to 5 February 2017, i.e. during the maximum of the minor stratospheric warming. Clearly the lidar and radiometer observations agree within their random errors. In contrast, all model and re-analysis data correspond well to the observations in the stratosphere but lie significantly outside the error bands above 0.3 hPa apparently failing to correctly represent the extent of the meospheric wind shear. With an offset of 17 m/s at 0.1 hPa MERRA2 is by far the closest to the observations while the offsets for ECMWF forecasts and SD-WACCM exceed

35 m/s. This could be an indication that the assimilation of mesospheric data from MLS drives the model closer to reality. The geostrophic wind computed from MLS disagreeing with the observations over the entire altitude range are due to the very localised effects in space and time during this minor warming being averaged out when considering the $5°/30°/24$ hour latitude/longitude/time window for these calculations.

## 6    Intercomparisons between near-continuous observations and models

Lidar observations are limited to clear sky conditions. Moreover particularly short lidar observations have not been considered in the observational intercomparisons for two main reasons: some minimal integration time is needed for guaranteeing a sufficiently homogeneous wind field of the horizontal area sampled by the instruments and for the wind radiometer to deliver stable wind retrievals. Therefore the results in Sect. 5 only cover a comparatively small subset of the observations collected by WIRA. In the present section we aim to exploit the entire data set obtained by the wind radiometer at ALOMAR during the

study period, which almost covers a full annual cycle, and compare it with meteor radar, model and geostrophic wind data.

The time series of zonal and meridional wind profiles are shown in Figs. 9 and 10 along with bi-monthly average profiles of this data set in Figs. 11 and 12. The extent of the wind radiometer's error bars in Figs. 11 and 12 depends on the number of contributing measurement cycles at the respective altitude and on the signal-to-noise ratio of the wind signature in the recorded

spectra. The latter is mainly determined by the opacity of the troposphere at the observation frequency which is influenced by its variable liquid water and water vapour content. Similarly the measurement conditions influence the upper limit of WIRA's trustworthy altitude range. In Figs. 11 and 12 USLM data at each altitude are only considered when the radiometer observations are judghed trustworthy at this level. This guarantees that all USLM average profiles are based on simultaneous observations/data. As this approach is not possible for the non-overlapping altitude range of the meteor radar, its profiles are averages over all days. This may lead to slightly different temporal sampling between the USLM and the meteor radar data for the tree panels of the summer half-year when WIRA's uppermost trustworthy altitude is not constantly adjacent to the 0.02 hPa line (see Figs. 9 and 10). Moreover, it should be noted that in contrast to the USLM observations meteor radar winds are never convolved with WIRA's averaging kernels according to Eq. (2).

The ECMWF panel in Figs. 9 and 10 is shared by ERA5 before 31 December 2016 and forecast data afterwards. For a full intercomparison ECMWF's ERA5, its forecast and its operational analysis winds between 1 August and 31 December 2016 are reprinted in the Supplement's Fig S5 which confirms that they follow a very similar pattern. Notable differences between ECMWF forecast and ERA5 are, however, found in the zonal and meridional winds above 0.2 hPa for October - November 2016 (Figs. 11 and 12) which according to Fig. S5 are not related to the change of model cycle in the ECMWF forecasts on 22 November (before this date ECMWF forecast and ERA5 both ran on 41r2), but rather to ERA5 systematically featuring lower absolute zonal and meridional wind speeds at these altitudes.

The fundamental pattern of the time series in Figs. 9 and 10 is the same for all model data and the radiometer observations. Similarly, in cases where the uppermost altitude of trustworthy wind radiometer data and the lowermost level of meteor radar observations are adjacent, the hand-over of the wind profile between these instruments is remarkably smooth without major jumps in the wind profiles. This behaviour is especially well illustrated for the rapidly changing meridional winds but it can also be discerned for the zonal component.

Despite the good overall agreement of the middle-atmospheric data sources, some differences can be distinguished. For instance, the mesospheric westward wind above 0.2 hPa is substantially weaker in the ERA5 re-analysis and the ECMWF forecasts in comparison to WIRA, MERRA2 and the geostrophic wind computations from MLS in August 2016 which translates to a substantial bias in the upper part of the upper left panel of Fig. 11. SD-WACCM also features comparable wind speeds as the radiometer observations in August, but the strong mesospheric eastward winds around 28 Aug and 6 Sep, which could be artefacts, drive the bi-monthly average towards zero. Therefrom one might conclude that the advantages of high altitude input data (MERRA2 and MLS geostrophic wind) or high model tops (SD-WACCM) drive these models closer to the observations than ECMWF.

In the same panel stronger eastward winds of WIRA with respect to all comparison data can be distinguished below 0.4 hPa where they lie slightly outside the error bands. This is most probably related to the strong eastward winds measured by the radiometer on several consecutive days at the end of August which are not seen in this strength by any other source of comparison data. The reason for this difference could not be established and unluckily there are no lidar measurements at this time which could provide independent observational evidence.

SD-WACCM features substantially lower zonal wind speeds compared to all data sets and especially to the observations for October/November and December/January above 0.1 hPa. For Dec/Jan this tendency seems to be confirmed by the meteor radar wind whereas these are almost equally offset from the high-altitude radiometer and SD-WACCM data for Oct/Nov.

Finally, for February/March 2017 ECMWF forecasts, SD-WACCM and radiometer measurements are in close correspon-
dence while MLS and the geostrophic winds have offsets of up to 7 m/s around 0.1 hPa. Besides these few exceptions the agreement of the zonal wind from the different data sources in Fig. 11 is very good and the comparison data often lie within the error bars of the radiometer.

Regarding the meridional component, winds from all data sources agree with each other and are generally within the ob-
servation errors (Fig. 12) for almost all bi-monthly average periods. The only notable exceptions occur for SD-WACCM in
April/May 2017 with offsets of up to 10 m/s and, more pronounced, in February/March 2017 when all comparison data sug-
gest higher winds than observed by the radiometer above 0.2 hPa. Again, the spread between the model data is considerable with offsets between WIRA and ECMWF extending from 7 m/s at 0.1 hPa up to 12 m/s at 0.3 hPa while the difference between the observations and SD-WACCM remains below 3 m/s at all altitudes. In both panels the transition from the radiometer to the meteor radar wind profile is almost perfectly smooth justifying some trust in the radiometer observations. The difference
pattern for February/March 2017 is obviously a related feature to what has been observed during the lidar intercomparisons in Fig. 7. When focusing on this period in the time series in Fig. 10 it appears that the two episodes of strong northward wind at the beginning and end of February extend to much higher altitudes in ECMWF than in WIRA data. A similar but reduced tendency is visible for MERRA2. However, the meteor radar observations during these periods seem to confirm the lower velocity meridional winds seen by the radiometer at high altitudes by also showing low adjacent velocities above 0.02 hPa.

In addition to the previously discussed long-term comparisons, interesting short-term events can be distinguished from the time series. One example shall briefly be discussed here: On 15 and 16 January 2017 a reversal of the zonal wind direction in the mesosphere is clearly visible in the microwave radiometer observations in Fig. 9 while the stratospheric winds remain at high eastward velocities. It appears that the transition from the WIRA to the meteor radar profiles is smooth also on these days with the meteor radar winds at the lowest levels also being reverted to westward direction. Moreover, a lidar observation in the
night from 14 to 15 January corresponded very closely to the radiometer profile (see Fig. S3, second line/second column). Thus, there are good reasons to think of this inversion as a true atmospheric feature rather than a measurement artefact. MERRA2 and the geostrophic wind calculations from the MLS geopotential height feature a similar pattern as WIRA with a clear reversal of the zonal wind direction in the mesosphere. This change is not captured to its full extent by ECMWF's forecasts and SD-WACCM which only show a reduction in mesospheric wind speeds. Hence the feature is present in all data sets which are
either direct observations or use observations from mesospheric altitudes (MLS) as base for the calculations or as assimilation data. The fact that it is not seen in ECMWF which only assimilates a few infrared temperature data and SD-WACCM which is completely free-running in the mesosphere may be interpreted as an indication that this effect is not captured by the model physics and solely exists when real observations are considered.

## 7 Conclusions

Following the recent developments of the wind measurement techniques of Doppler microwave radiometry and lidar iodine absorption spectroscopy two such instruments have been operated in co-location at Andenes (69.3° N, 16.0° E) for a 11 months intercomparison period. After Lübken et al. (2016) had found good correspondence of nighttime lidar winds with 8 rocket soundings, the present study can be regarded as the first thorough cross-instrumental validation of the new lidar and radiometry techniques for wind observations in the upper stratosphere and lower mesosphere during night and day. This part of the study is based on 518 hours of coincident observations by the ALOMAR RMR lidar and the microwave radiometer WIRA with individual recordings having a minimal duration of 5 hours. The intercomparisons have demonstrated the quality of the new measurement techniques which appear to be largely bias-free.

The comparison of the wind observations during sunlit periods prove that the ALOMAR RMR lidar can overcome the additional challenges for daylight operation. On the other hand, the nighttime observations confirm that the adjustments to the retrievals presented in Rüfenacht and Kämpfer (2017) allow wind radiometry to obtain accurate results under both day and night conditions. Especially the nighttime zonal winds are in very close agreement with all compared data sources while some differences in the meridional component appear above 0.3 hPa. It should however be noted that the overall nighttime averages are largely dominated by 9 consecutive nights of measurements in February 2017 so that this feature may also be due to a short-term localised effect. During this period the model meridional winds were found to be equally scattered between the radiometer and the lidar profiles (ECMWF close to lidar, SD-WACCM close to WIRA, MERRA2 in between) above 0.3 hPa indicating no clear preference for either of the observations. Meanwhile the lowermost levels of meteor radar measurements closely correspond to the uppermost radiometer winds.

Except for the previously mentioned nighttime meridional winds above 0.3 hPa biases are mostly below 5 m/s and within the random errors of the observations and never exceed 10 m/s which is less than 10% of typical wind speeds for this altitude range. In addition to the good average agreement between lidar and radiometer it should be noted that also the temporal and altitude-dependent features in the time series correspond very well as discussed for the 187 hours of continuous lidar observations in February 2017.

In conclusion, the observational intercomparisons prove that middle-atmospheric winds from both instruments can be used as single validated standards when operated at different sites or as complements when in co-location. Indeed, Doppler radiometry for weather-independent continuous monitoring and lidar spectroscopy for high-resolution observations when conditions permit appear to be an ideal combination of measurement infrastructure.

The 11 months time series comparison of quasi-continuous data reveals that the transition from the highest WIRA levels to the lowermost radar recordings at around 0.02 hPa is smooth, especially also for the meridional winds in February and March 2017 where the largest discrepancies to the models exist. In general the agreement among the different investigated models and re-analyses and with the microwave radiometer observations is very good for zonal as well as for meridional wind. Nevertheless, examples of pronounced short-term discrepancies between all models and the agreeing radiometer and lidar measurements have been identified.

The most prominent long-term bias has been found in August 2016 when the westward wind speeds above 0.2 hPa are underestimated by ECMWF's forecast data and ERA5 re-analyses by up to 10 m/s with respect to the radiometer measurements, whereas MERRA2 is in close agreement. To elucidate this difference we aim to target wind radiometer observations to future summers and autumn equinox transitions at high or mid latitudes, a period which had previously been discounted in view of the rapidly changing winter dynamics.

*Data availability.* ERA5 as well as forecasts and operational analyses from ECMWF are available at https://www.ecmwf.int/en/forecasts/ datasets, while MERRA2 re-analysis data can be obtained from https://gmao.gsfc.nasa.gov/re-analysis/MERRA-2/data_access. The temperature profiles from MLS used for the calculation of the geostrophic wind field are available at https://mls.jpl.nasa.gov. The lidar, wind radiometer and meteor radar observations as well as the computations of geostrophic wind and the SD-WACCM simulations can be made available upon request.

## Appendix A:  Validation of mesopause region wind retrievals by WIRA against meteor radar

Rüfenacht and Kämpfer (2017) proposed to exploit the signals recorded by ground-based microwave radiometers operated at ozone emission frequencies to obtain wind information from the mesopause region. Thanks to the co-location of the wind radiometer WIRA with the Andenes meteor radar the reliability of this approach can be investigated.

It should be noted that microwave radiometry at these altitudes has some limitations: First of all, observations are only possible at times for which enough emitters (i.e. ozone molecules) are present. This is typically the case during nighttime so that at polar latitudes the retrieval of information about this altitudes is not possible during summer. Moreover, the exact altitude of the signal can not be determined by the effect of pressure broadening as the linewidth of the emission spectrum at such low pressures is largely dominated by Doppler broadening. This implies that, unlike in the USLM, it is impossible to distinguish signals from different altitudes by their spectral shape. Thus, the attribution of the retrieved wind information to a certain altitude becomes highly dependent on the accuracy of the vertical distribution of the mesopause region ozone in the a priori information. In contrast, meteor radars are specifically designed for observations in the mesopause region and are thus expected to deliver more reliable wind estimates.

Figure 13 presents the zonal and meridional wind observed by WIRA and the Andenes meteor radar during the winter months of 2016/17. It demonstrates that the mesopause region wind observations by WIRA follow a similar pattern as the meteor radar winds, especially when these are convolved with WIRA's averaging kernels according to Eq. (2). The convolving here not only accounts for the different altitude resolution of the wind radiometer, but, by doing so, also mitigates the effect of possibly inaccurate altitude attributions of the signal as here WIRA's vertical resolution is basically equivalent to the vertical extent of the secondary ozone maximum. From the validation example provided here it may be concluded that in the absence of dedicated co-located instruments for mesopause region wind measurements nighttime wind radiometer data can be used as a source of information for this altitude range.

*Competing interests.* The authors declare that they have no conflict of interest.

*Acknowledgements.* This work has been funded by the Swiss National Science Foundation grants P2BEP2-165383 and 200020-160048. Moreover the observational part has been supported by the European Union's Horizon 2020 Research and Innovation programme under grant agreement No. 653980 (ARISE2) and by the German Federal Ministry of Education and Research through the program Role Of The
5 Middle atmosphere In Climate (ROMIC) initiative GW-LCYCLE. We acknowledge ECMWF for the ERA5, the forecasts and the operational analysis data, NASA for the MERRA2 and the Aura MLS data as well as the NCAR CESM working group for providing the SD-WACCM model code.

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

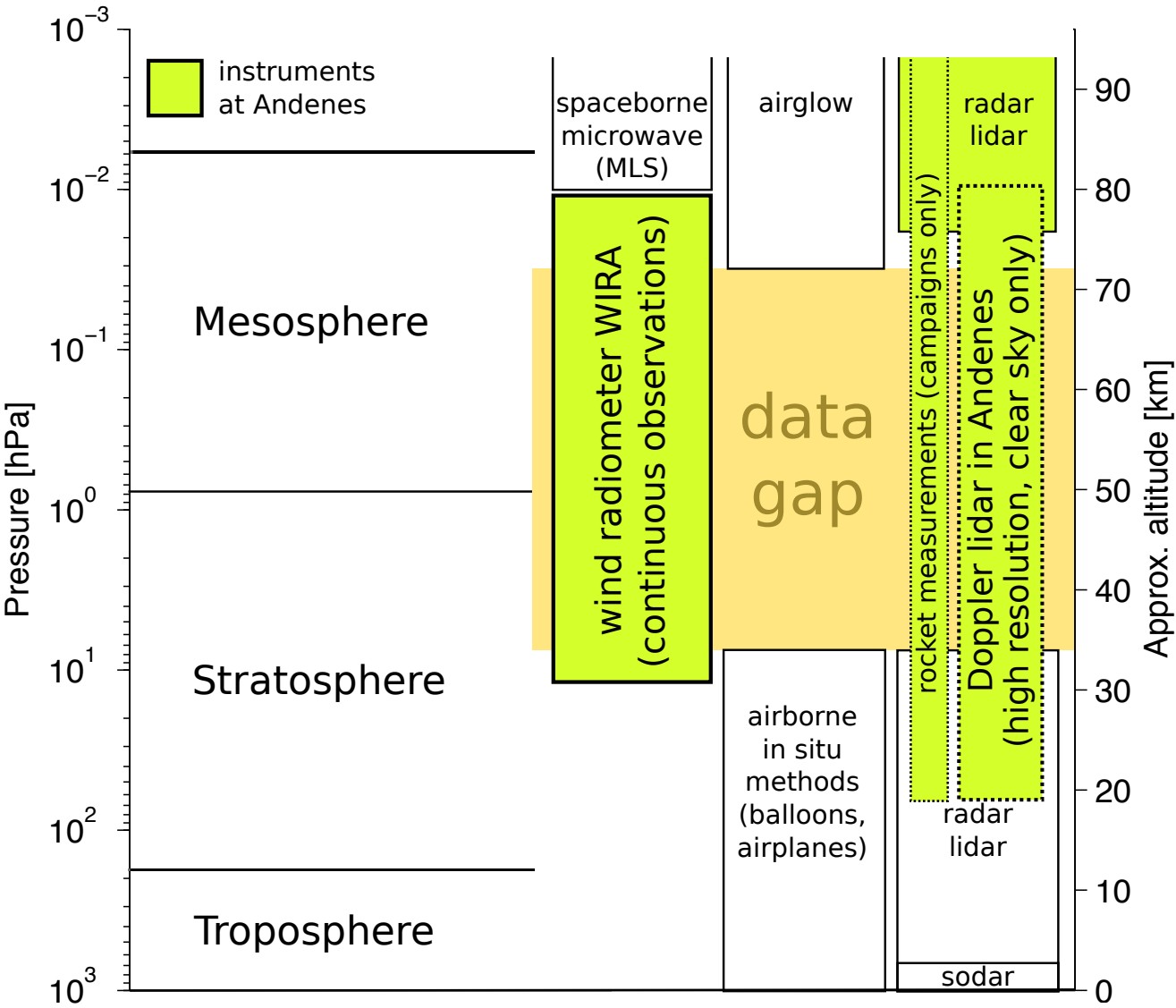

**Figure 1.** Overview of the altitude coverage of the currently operational wind measurement techniques. The techniques which are available at Andenes (69.3° N, 16.0° E), the observation site of the present study, are highlighted.

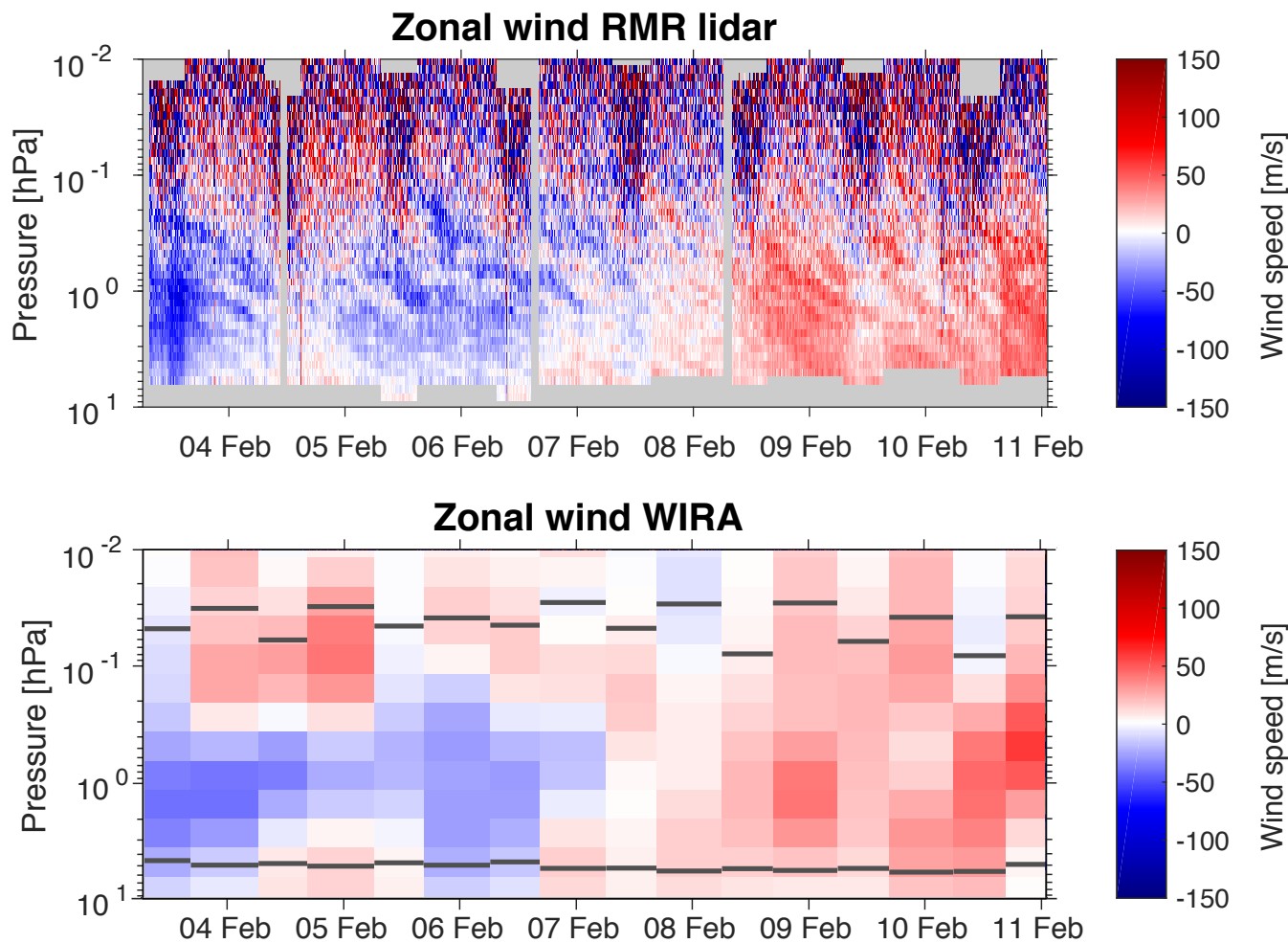

**Figure 2.** Time series of zonal wind during 187.1 hours of coincident lidar and radiometer measurements recorded at Andenes from 3 to 11 February 2017. For better visibility the lidar data are binned to 1 km vertical resolution while the temporal resolution is 5 minutes. The trustworthy altitude range of the wind radiometer data, according to the definition given in Sect. 2.1, is marked by the horizontal dark grey lines. Data outside this range should not be considered as it may substantially be affected by a priori assumptions. All times are expressed in Coordinated Universal Time (UTC) with the ticks at 00:00.

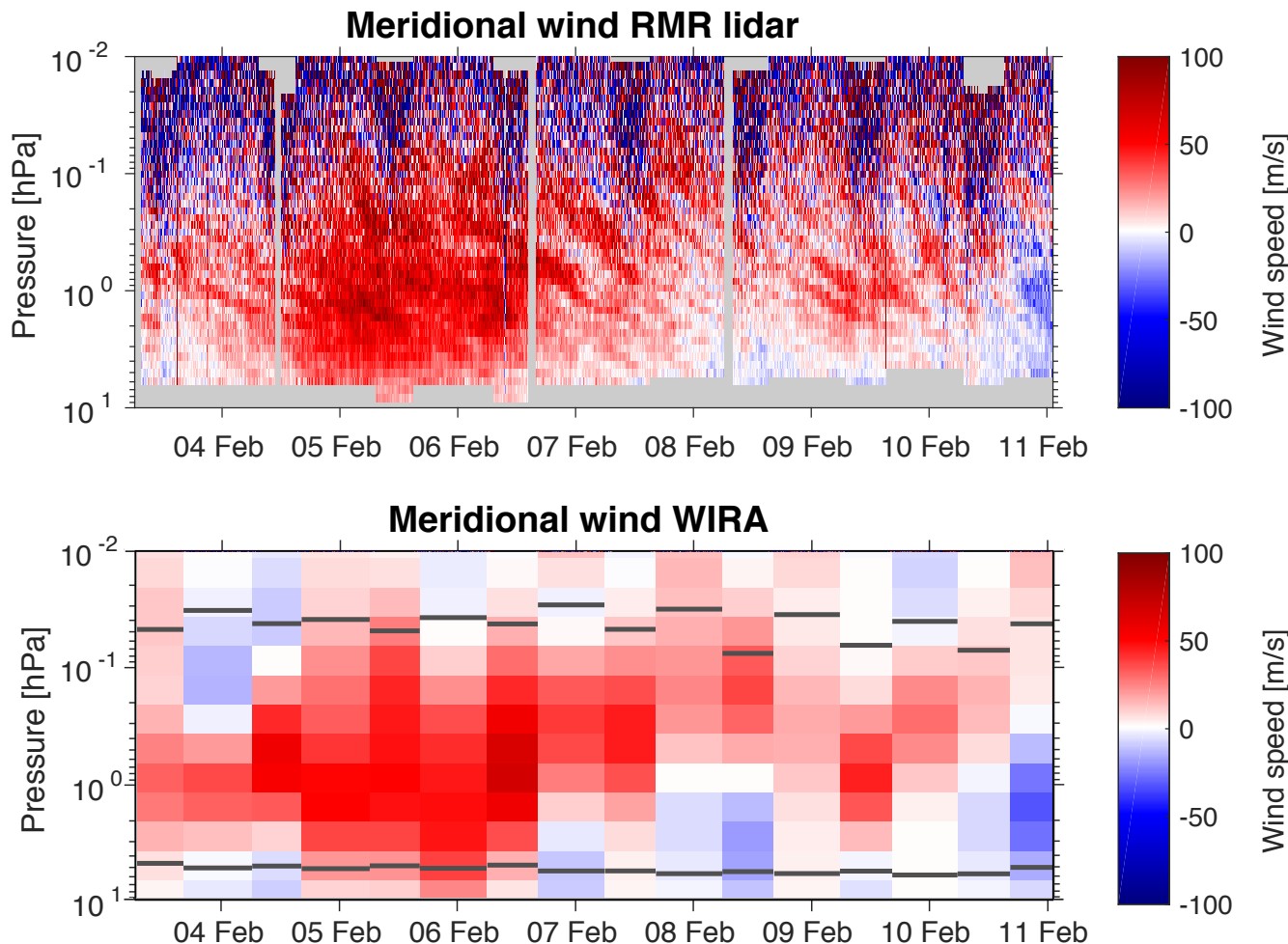

**Figure 3.** As Fig. 2 but for the meridional wind component.

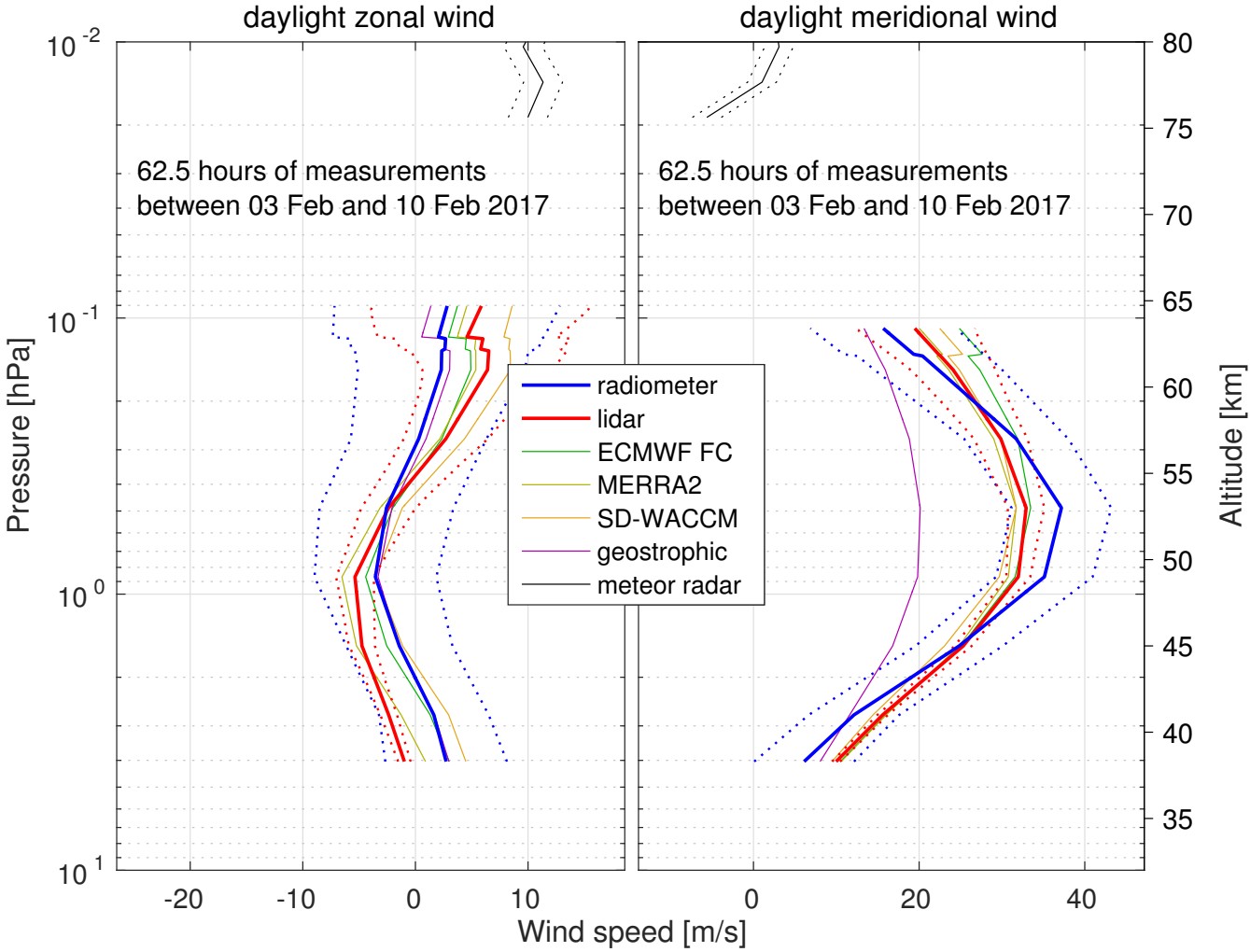

**Figure 4.** Mean daytime zonal and meridional wind profiles for the observation period 3-11 February 2017 measrured by the wind radiometer WIRA and the RMR lidar in comparison with models, re-analyses and geostrophic wind calculations from MLS geopotential heights. All middle-atmospheric comparison data have been convolved with WIRA's averaging kernels according to Eq. (2) and cut at the limits of the trustworthy altitude range of the coincident radiometer observation. The discontinuities in the profiles visible at the uppermost altitudes originate from the averaging over observations with slightly different altitude coverage. At the uppermost altitudes the simultaneous observations from the meteor radar (not convolved) are shown. The random errors of all measurement data are denoted by dotted lines. Only data from times when lidar and radiometer were both operated in their respective daylight mode have been considered in this plot.

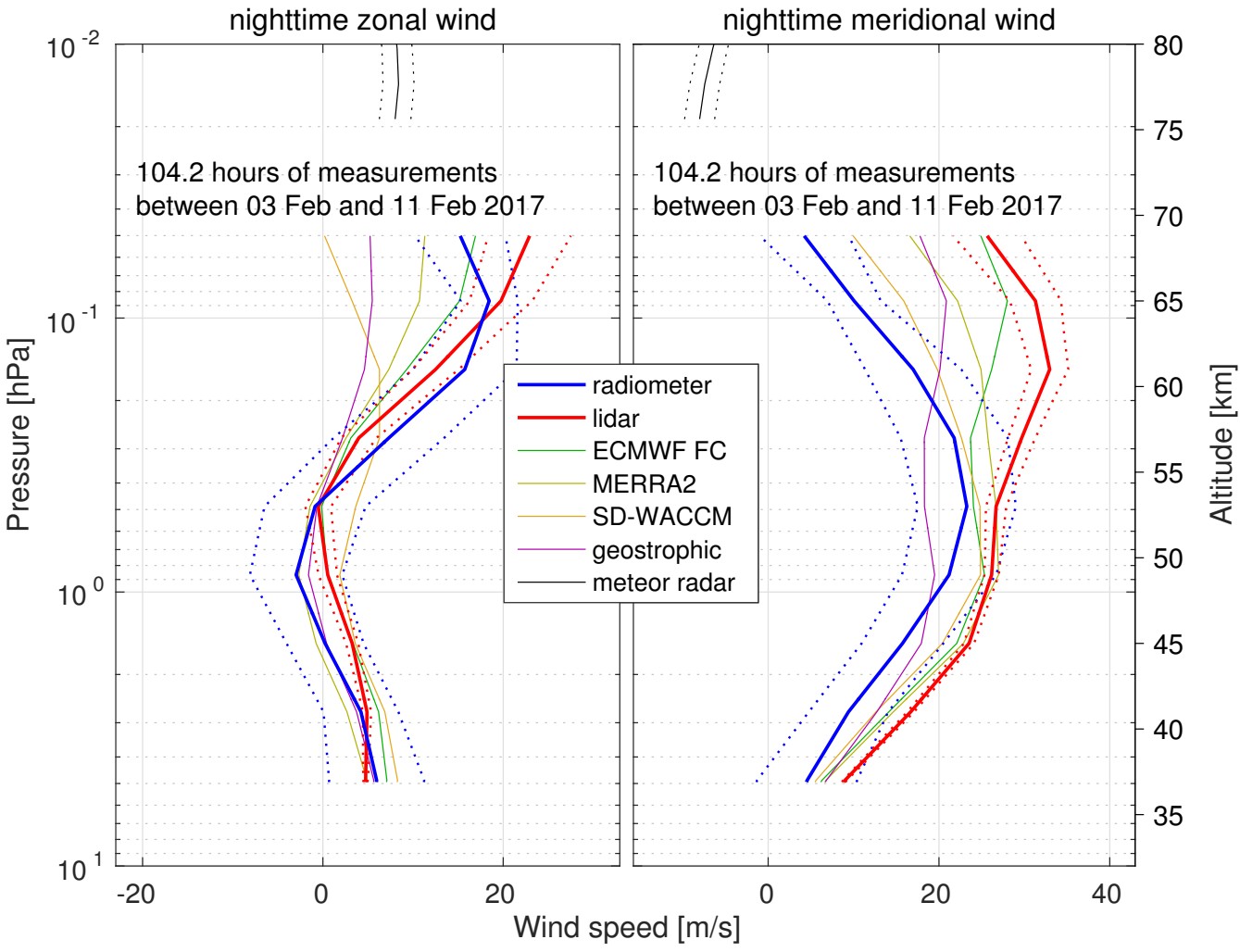

**Figure 5.** As Fig. 4 but for data from the time period when both instruments were operated in their respective night mode.

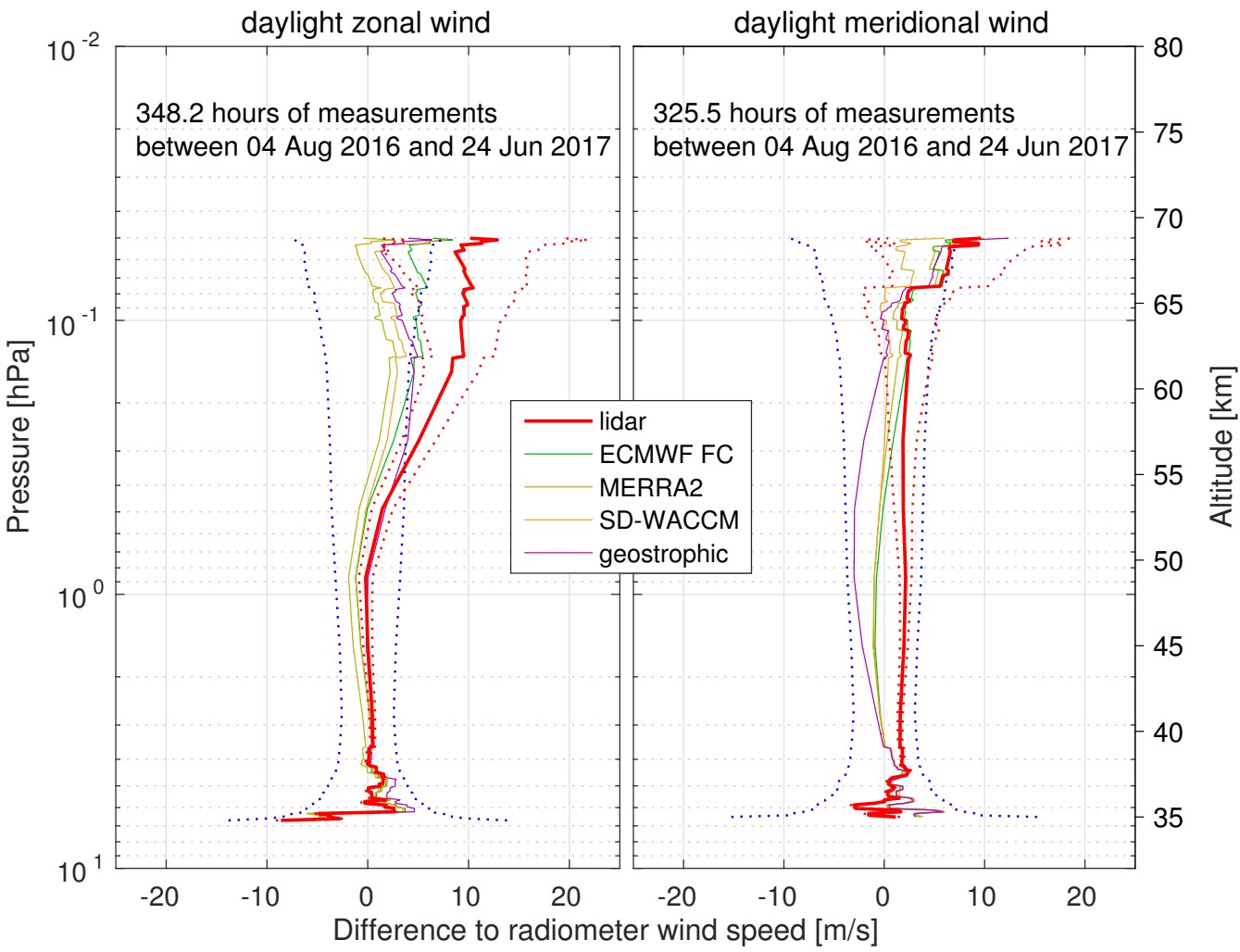

**Figure 6.** Mean difference of all convolved daytime zonal and meridional wind profiles from models, re-analyses, MLS geostrophic winds and the RMR lidar to the observations from WIRA recorded during the entire intercomparison campaign. The bands of random errors of the radiometer and the lidar observations are denoted by blue and red dotted lines, respectively. The discontinuities in the profiles visible at the upper and lowermost altitudes originate from the averaging over observations with different altitude coverage. Only data from the time period when both instruments were operated in their respective daylight mode have been considered.

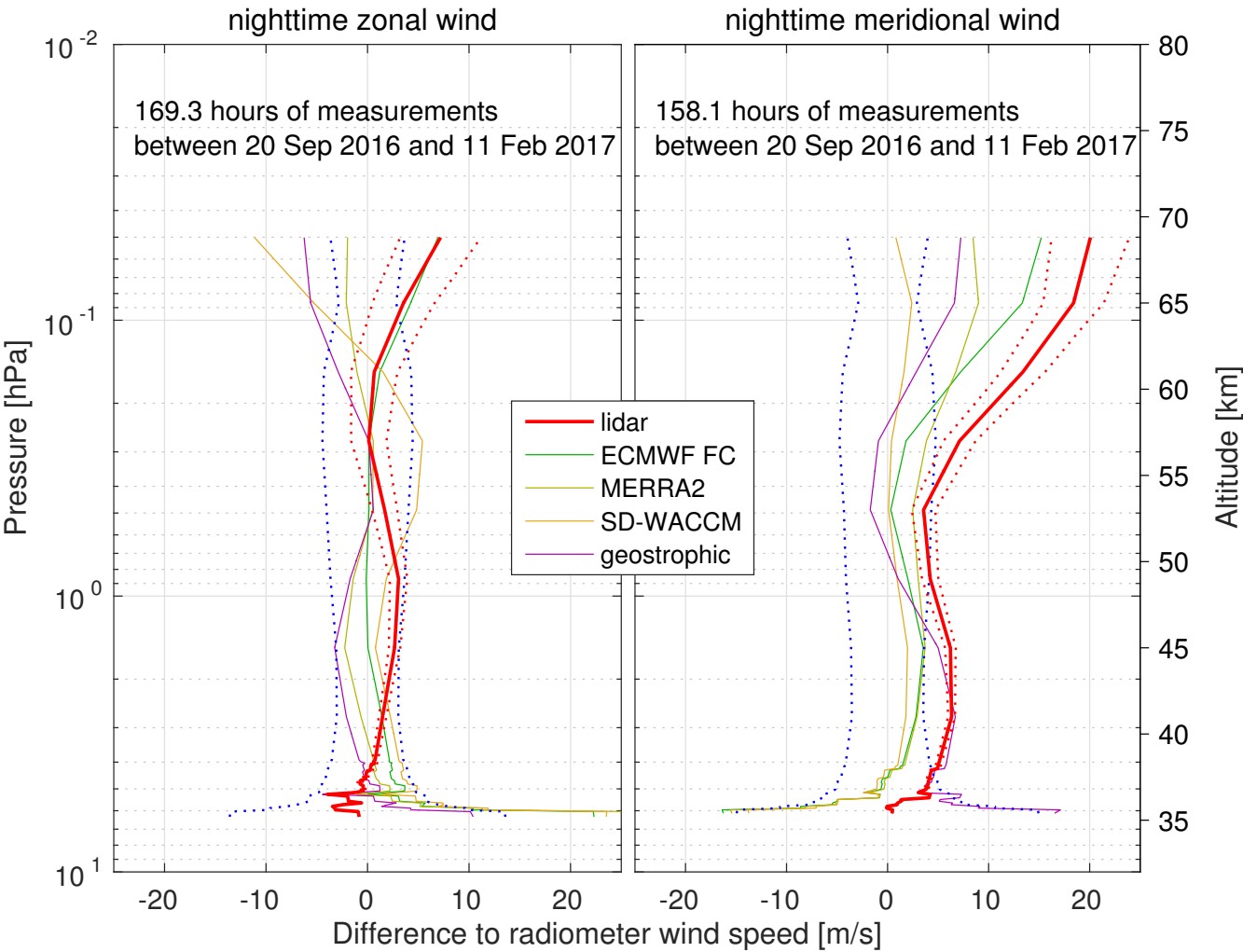

**Figure 7.** As Fig. 6 but for data from the time period when both instruments were operated in their respective night mode.

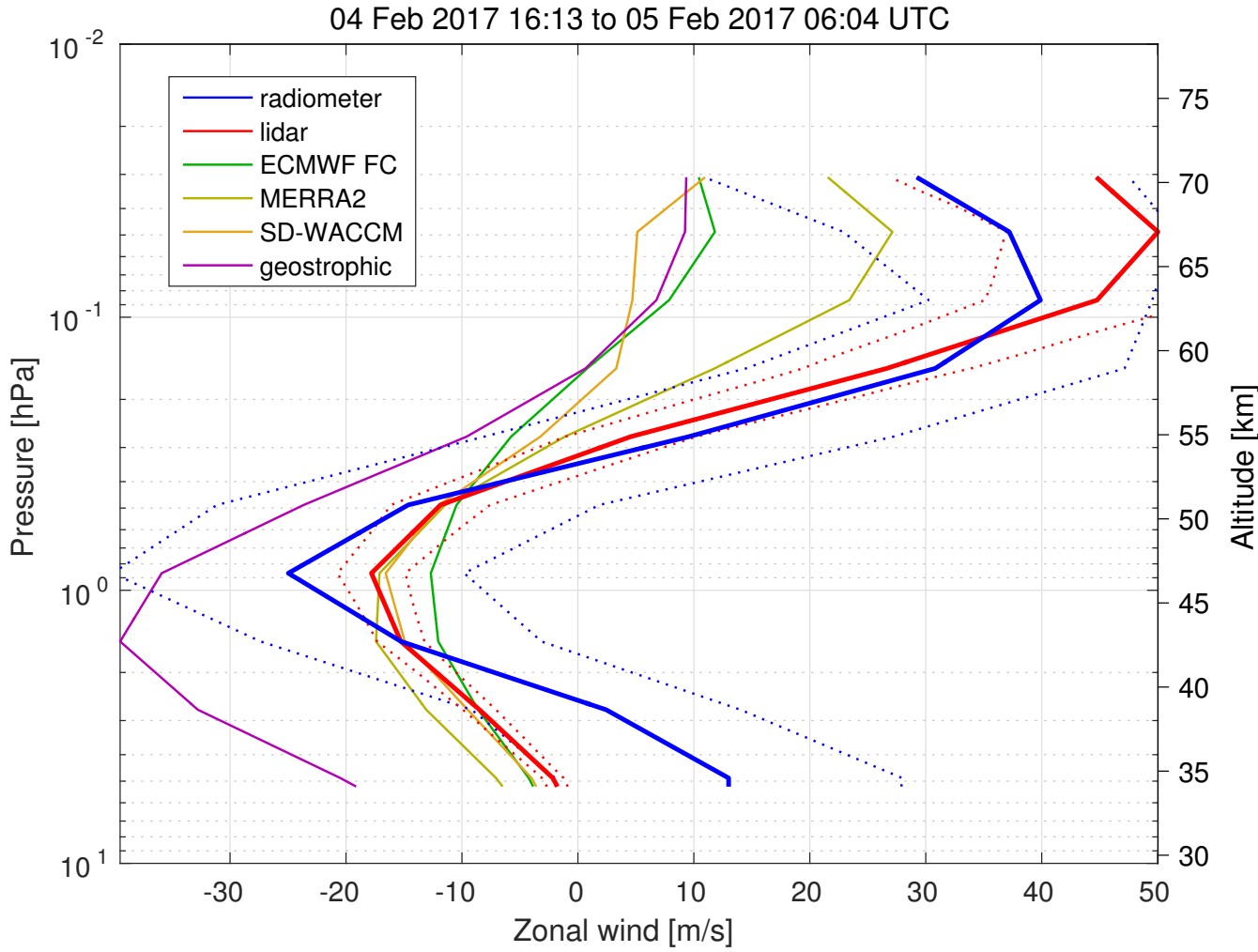

**Figure 8.** Mean profile of zonal wind for the part of the night from 4 to 5 February 2017 when both measurement systems were operated in night mode. For reasons of clarity only profiles convolved according to Eq. (2) are shown here, the same plot including the unconvolved wind profiles can be found in the Supplement's Fig. S3 (second row, fourth column). The blue and red dotted lines represent the error bands of the radiometer and lidar observations.

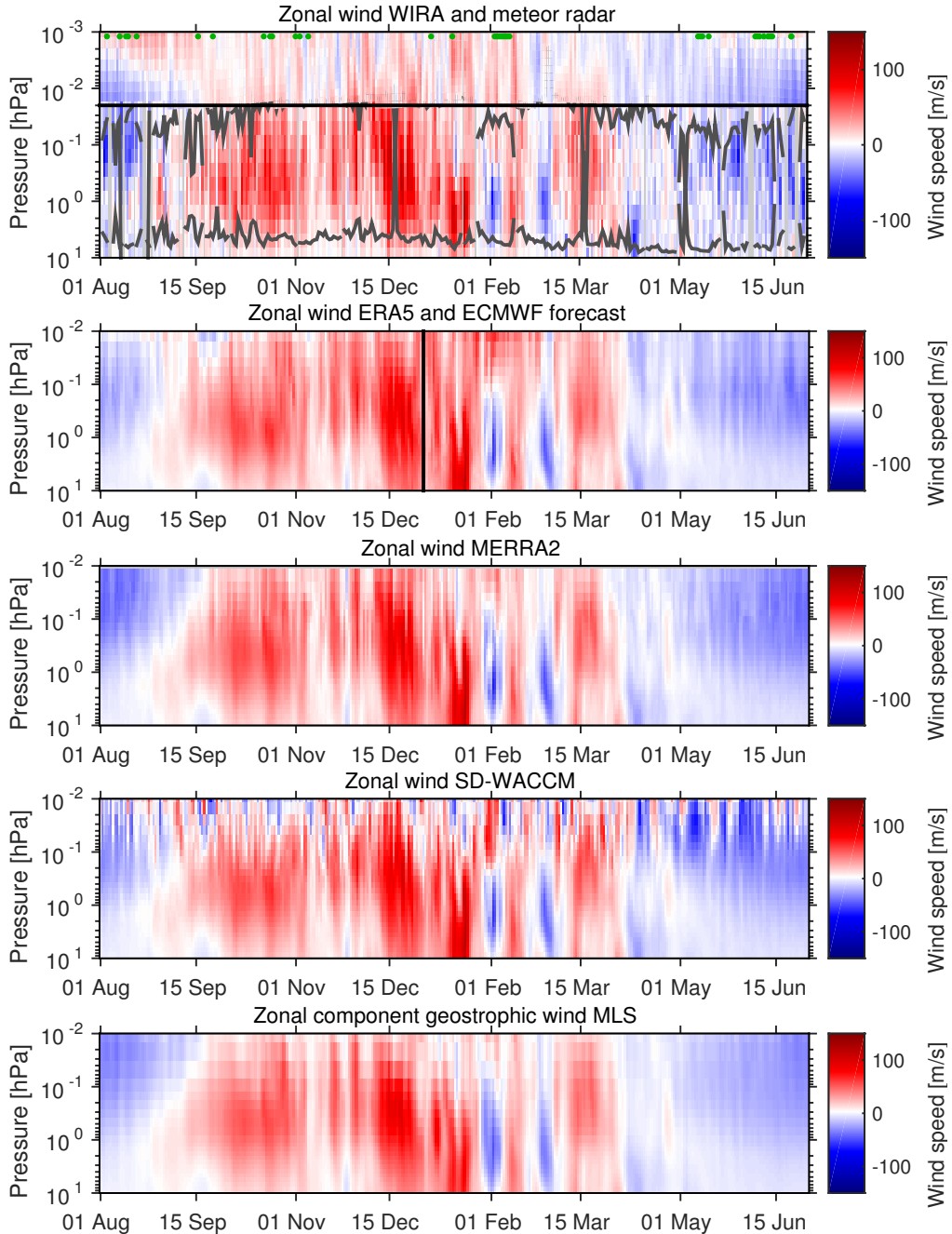

**Figure 9.** Time series of continuous or near-continuous zonal wind data at Andenes for the time period between 1 August 2016 and 30 June 2017. The green dots in the uppermost panel mark the dates where wind measurements from the RMR lidar are available. The uppermost panel shows wind radiometer data below the horizontal black line and meteor radar data above. The dark grey lines in the radiometer data denote the altitude limits within which WIRA data are trustworthy according to the conditions stated in Sect. 2.1. Radiometer data beyond this range are noticeably influenced by a priori assumptions should not be used for comparisons e.g. with meteor radar observations. At the time of writing ERA5 was only available until 31 Dec 2016 (black vertical line), therefore ECMWF forecast data is plotted for 2017.

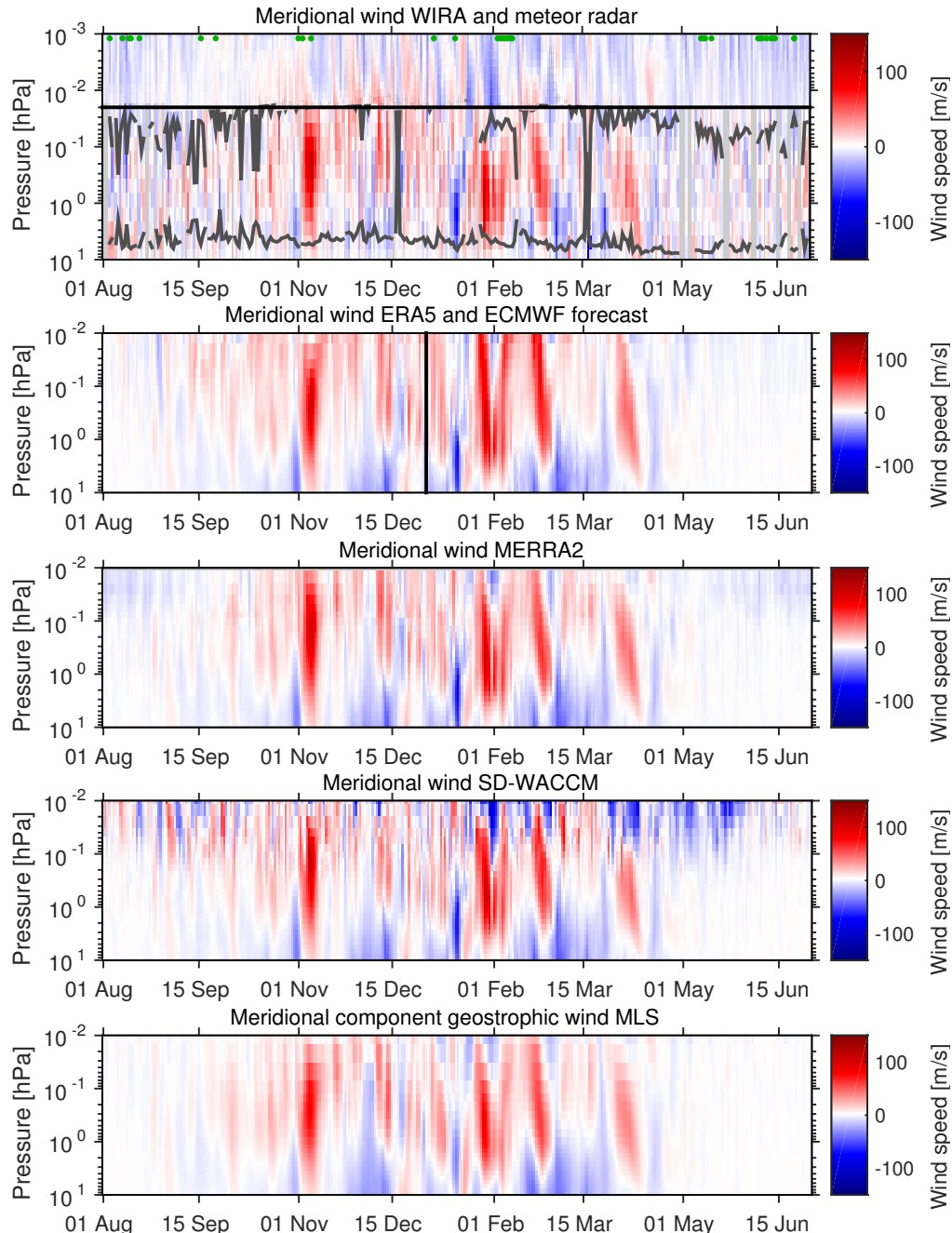

**Figure 10.** As Fig. 9 but for the meridional wind component.

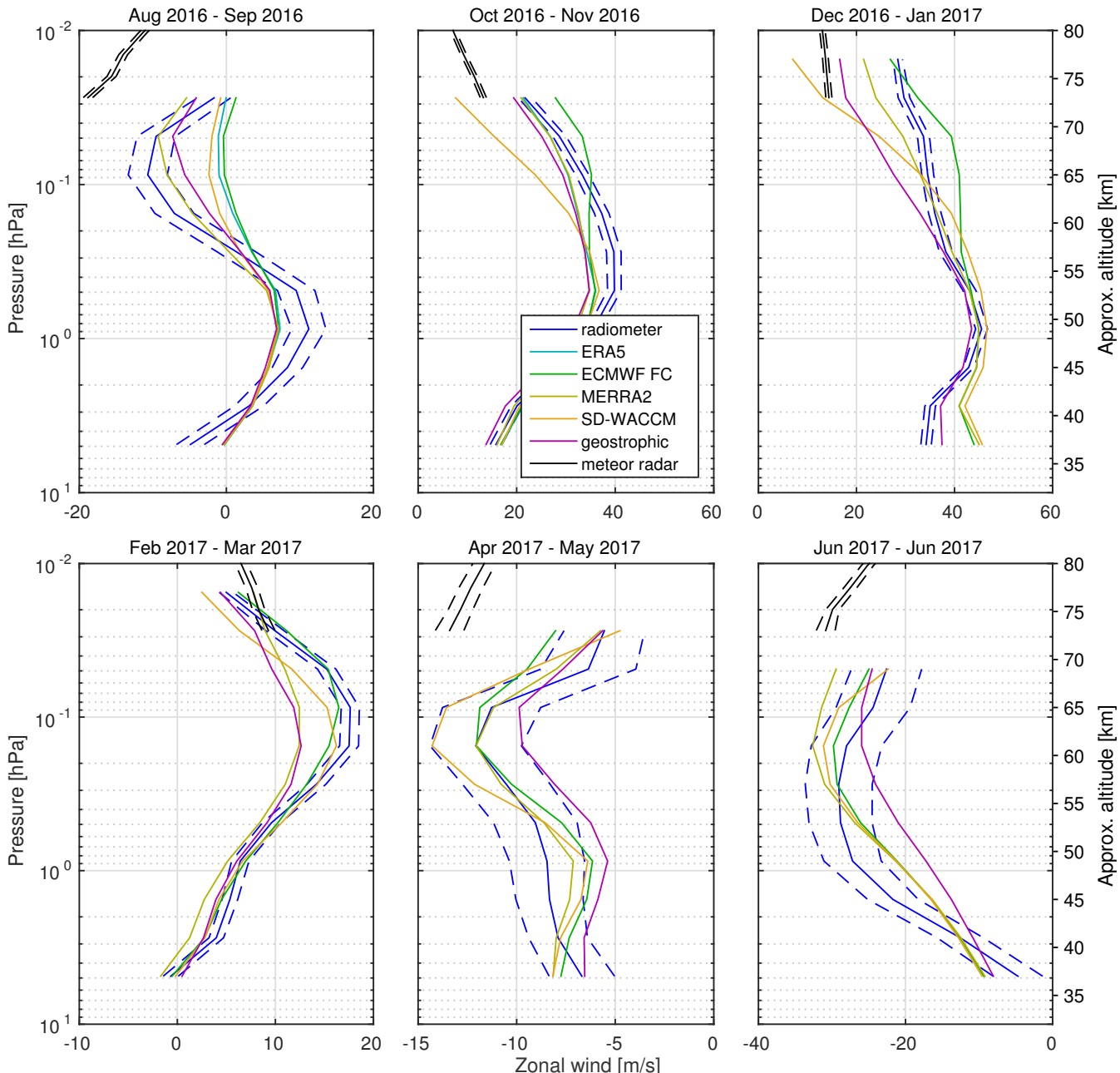

**Figure 11.** Bi-monthly average profiles of zonal wind at Andenes from re-analyses and models convolved according to Eq. (2) in comparison with observations from the wind radiometer WIRA and its random error (dashed). ERA5 data were only available for the first two panels at the time of writing. At the uppermost altitudes the raw, i.e. unconvolved, meteor radar wind profiles are shown. Due to the temporal variations of the upper altitude limit of the radiometer observations visible in Figs. 9 and 10 the sampling period of the meteor radar average wind can be rather different from the highest levels of middle-atmospheric data especially for the summer half-year, i.e. in the upper left, the lower centre and the lower right panel.

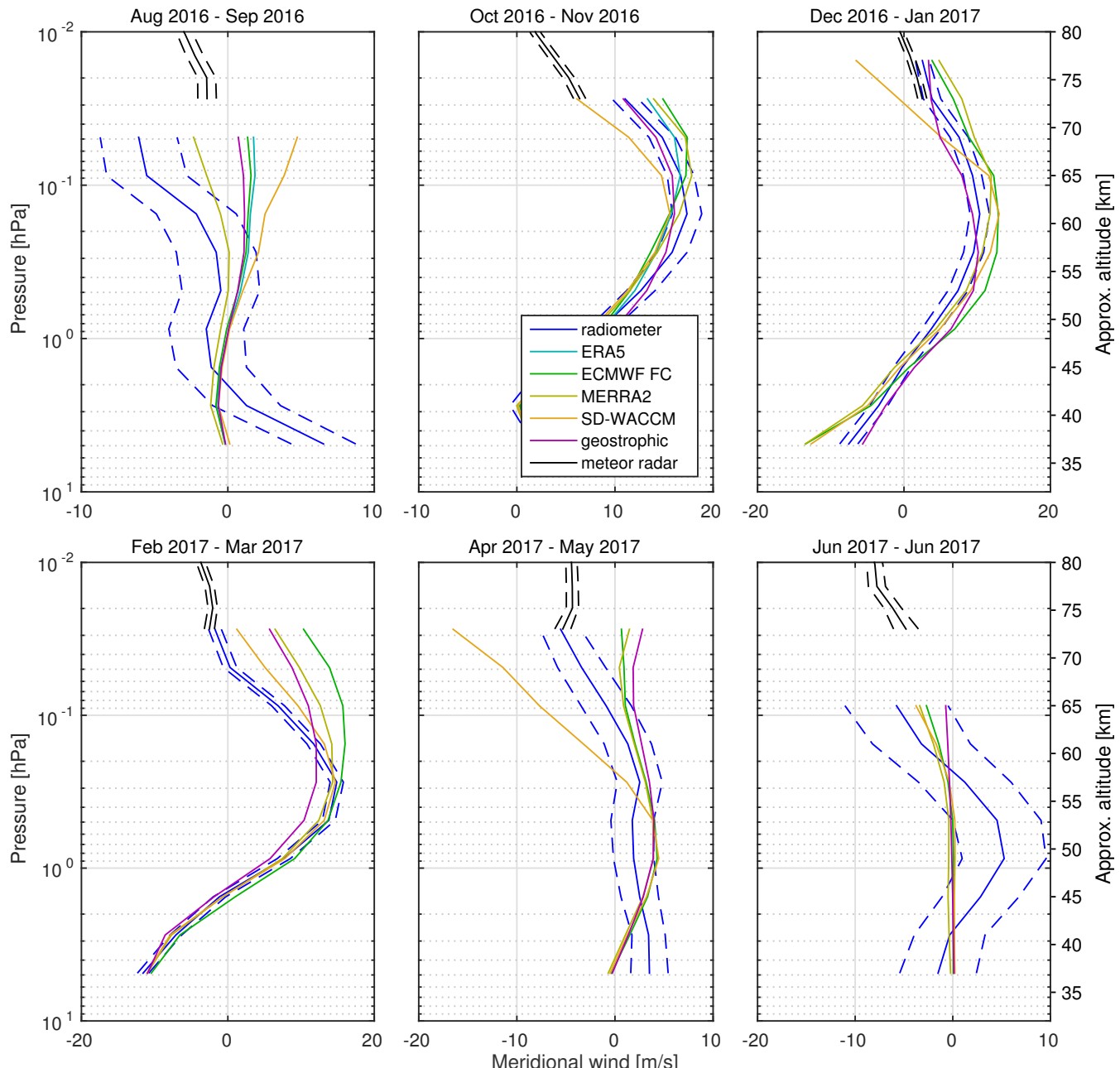

**Figure 12.** As Fig. 11 but for the meridional wind component.

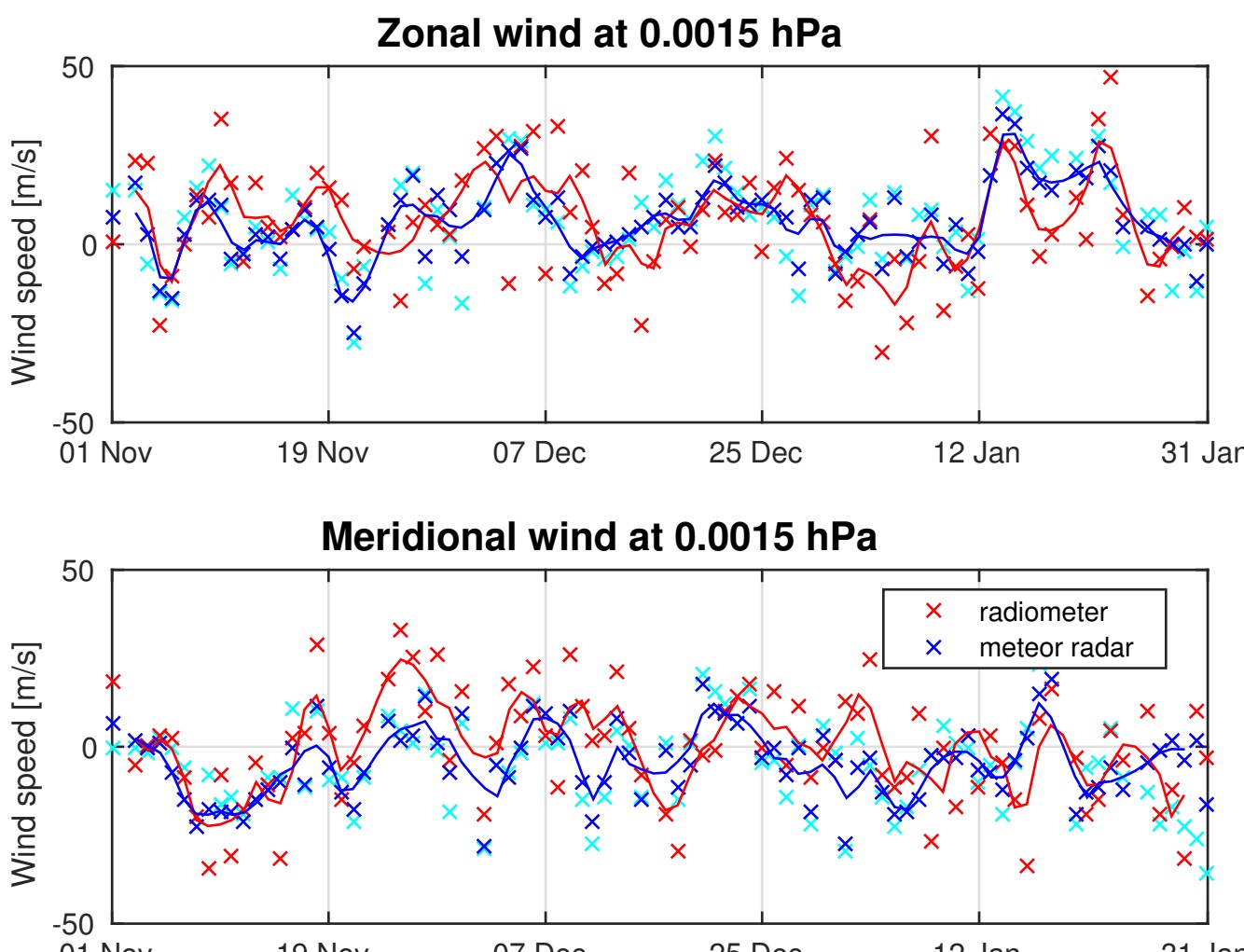

**Figure 13.** Observations of nighttime zonal and meridional wind in the mesopause region from the wind radiometer WIRA (red) and the Andenes meteor radar (dark blue: convolved with WIRA's averaging kernels according to Eq. (2); cyan: raw) for the winter months of 2016/17. The crosses denote the nightly averages whereas the coloured lines show a smoothed version of the data by a moving average filter with a span of 5 days.