# Peer review of "Validation of middle-atmospheric wind in observations and models"

_Atmospheric Measurement Techniques, 2017_

## Referee Comment (RC1) · Anonymous Referee #1 · 21 Dec 2017

This manuscript presents inter-comparisons between a radiometer capable of horizontal wind measurements in the middle atmosphere (WIRA) and a lidar (ALOMAR RMR) capable of wind measurements in addition to temperature and aerosol properties. Inter-comparisons are also made between these measurements and the ALOMAR meteor radar (although these measurements do not cover the same height region), as well as various models/re-analysis data sets. It is a useful study highlighting the capabilities of both the WIRA and the ALOMAR RMR to make useful measurements of wind in the upper stratosphere and lower mesosphere, a particularly difficult region of the atmosphere to measure wind in.

In its current form the manuscript reads as three separate studies with a common linkage, the WIRA instrument, which is compared to the lidar and radar measurements

and to various re-analysis data sets. Hence, I suggested the title change and in general suggest the Abstract, Introduction, and Conclusion focus on the validation of WIRA as the "hub" of the study, as for instance, there are not enough lidar measurements to make this a validation between WIRA and the lidar and poor/no overlap with the meteor radar.

I have the following suggestions for the authors to consider, which I think will improve the manuscript.

1. Title. Currently: "Validation of middle-atmospheric wind in observations and models" does not read well, nor adequately describe the study. My suggestion: Validation of Microwave Radiometer Wind Measurements Using Active Remote Sensing and Models

2. I am not an expert in microwave techniques. That said I am uncomfortable with a new correction to the wind retrieval involving being introduced (a correction for mesospheric ozone) via a short description in an Appendix. The authors should write a more detailed manuscript on this improvement to the technique, including demonstrating the affect of various realistic mesospheric ozone profiles on their original and revised technique, as well as validation of the improvement by comparisons with measurements if possible. It is OK to describe the ozone correction in the text and apply it to the measurements, I don't think this paper should be a justification/validation of this correction since you are not specifically picking out examples and showing improvements in the results. Comparisons with/without this correction are important, but I see them as beyond the scope of this paper but requiring a more careful assessment/validation.

3. Section 2.1 Spatial averaging in WIRA could explain the poor meridional agreement in the mesosphere with the lidar. Changes in meridional flow, particularly at high latitudes, can be abrupt in latitude; the larger latitudinal spread of WIRA to determine a wind could be averaging across two very different flows, while the lidar is sampling the same one.

4. Section 3.4 Limitations of the geostrophic analysis should be discussed a bit further,

pointing out the affect of curvature of the isobars and of baroclinic instability on the assumption of geostrophic balance.

5. End of Section 3: a semi-diurnal tide might average out but would a diurnal tide? What about tidal/mean wind/planetary wave interactions? Tidal structures can be complicated at higher latitudes. Please give a more complete assessment of these affects in the manuscript.

6. To continue on the previous point, the large differences between the meridional wind a greater heights could be due to Points 5 & 6. Please discuss this possibility.

7. Section 4.2: you discuss random uncertainties, but what of the systematic uncertainties? Systematic uncertainties can have a large affect on a wind measurement.

8. Section 6 - more discussion of the poor agreement between WIRA and the meteor meridional radar wind is needed, the wind variations on either side of the "line" in Figures 9 and 10 are huge. Figure 13 attempts to argue this isn't so bad, but is not called out in the main body of the text, and no detailed explanation of what the "convolved version" of the measurements means.

9. Some modifications to the conclusion are suggested, and are included on the marked up copy of the manuscript.

10. While I agree there is not an over-abundance of Rayleigh lidar wind measurements, there have been numerous papers by groups in France, the United States, and others since the 1980s (e.g. Chanin, Tepley, Meriwether, Keckhut). I believe there is also some comparisons between the lidars at La Reunion and the WIRA instrument? I suggest you mention some of this previous work in the Introduction.

11. The review of the capabilities of radars in this region is incomplete and should be revised. Some VHF radars can measurement wind below 70 km, but this capability is latitudinally dependent. The Japanese Antarctic radar, Pansy, has made wind measurements down to 55 km. MF radars routinely get measurements down to 60 km. I would

suggest a paragraph explaining what the radars can and can't do, and how WIRA and the lidar measurements complement the radar work. Please see the recent book by Hocking et al, which gives many of these references (Hocking, W., Röttger, J., Palmer, R., Sato, T., & Chilson, P. (2016). Atmospheric Radar: Application and Science of MST Radars in the Earth's Mesosphere, Stratosphere, Troposphere, and Weakly Ionized Regions. Cambridge: Cambridge University Press. doi:10.1017/9781316556115).

Other minor suggestions and changes are indicated on the marked-up copy of the manuscript attached.

Please also note the supplement to this comment:
https://www.atmos-meas-tech-discuss.net/amt-2017-390/amt-2017-390-RC1-supplement.zip

———————————————————

---

## Referee Comment (RC2) · Anonymous Referee #2 · 27 Dec 2017

This work concerns the comparison of wind measurements using microwave and lidar. This work is important while very few wind measurements are available and observations, while difficult to evaluable, are essential for model evaluations. This work if pertinent for AMT.

Minor comments Page 1, authors should refer to the background European project for such inter-comparison and should give some refrences about the project Page 1 line 23 the region USLM is not often used, I suggest the use of Middle Atmosphere Page 2 line 17 It concerns not only wind but temperature. Refer to Le Pichon et al. Page 2 section 2 No reference is performed on Radiosondes. Figure 1 suggest that no RS exist at Andenes. Is it true?

Figure 9 and 10 Wind differences are difficult to estimate from these figures. Is it

possible to provide differences instead of wind profiles?

---

## Author Comment (AC2)

**Reply to anonymous referee #2**

Review of doi: 10.5194/amt-2017-390

R. Rüfenacht et al. 2017

- blue: reviewer comments
- green: author replies

**General comments:**

- This work concerns the comparison of wind measurements using microwave and lidar. This work is important while very few wind measurements are available and observations, while difficult to evaluable, are essential for model evaluations. This work if pertinent for AMT.

**Minor comments:**

- Page 1: authors should refer to the background European project for such inter-comparison and should give some references about the project
We extended the the introduction in this sense: "... are at a very early stage. Such intercomparisons at multi-instrument sites are a key activity of the Horizon 2020 project ARISE[1] (Blanc et al., 2017). Previously, Lübken et al. (2016) presented..." with the foothonte "[1]http://arise-project.eu". We also mentioned this project at the beginning of Sect. 2: "Wind radiometer, lidar and meteor radar are all contributing to the before-mentioned ARISE project (Blanc et al., 2017)."

- Page 1 line 23: the region USLM is not often used, I suggest the use of Middle Atmosphere
We intentionally used the terminology upper stratosphere lower mesosphere (USLM) in this paper because it is more precise than middle-atmosphere. Indeed the core observations by the lidar and the radiometer do not cover the entire middle-atmosphere but rather the USLM which is also the altitude range where the gap of observations is present. In the lower stratosphere (which is included in the term middle atmosphere) wind a whole network of wind observations e.g. by radiosondes and radar wind profilers exists whereas the mesopause region and the uppermost part of the mesosphere can be assessed by different types of radar. The term USLM is also used by other authors (e.g. Baron et al. 2013). For this reasons we prefer to stick to the terminology USLM. We checked that the abbreviation is explained at the first occurrence in the abstract and the introduction

- Page 2 line 17 It concerns not only wind but temperature. Refer to Le Pichon

et al.

We agree that observational validations of middle-atmospheric temperature are also rare although MLS is providing global observations of temperature throughout the entire range between approximately 16 and 90 km. As we do not want to confuse the reader and make clear that the focus of the present paper lies on wind we added the reference to Le Pichon et al. in form of a footnote: "Le Pichon et al. (2015) noted that also middle-atmopsheric temperature is a little-validated product."

• Page 2 section 2 No reference is performed on Radiosondes. Figure 1 suggest that no RS exist at Andenes. Is it true?

It is true that there are no routine radiosoundings at Andenes. Some soundings have been performed on campaign basis, but not during the time period under investigation. This does however not negatively impact the present study because radiosondes generally reach top altitudes of 30-35 km and would thus not provide significant altitude overlap with the lidar or the wind radiometer anyway. The major general circulation models broadly assimilate radiosoundings and are validated against observations at these altitudes so that this is not the focus of the present study.

• Figure 9 and 10 Wind differences are difficult to estimate from these figures. Is it possible to provide differences instead of wind profiles?

We completely agree that it is hard to draw quantitative comparisons from Figs. 9 and 10. The primary aim of these figures is to present the time series showing how the dynamical patterns and their temporal evolution are captured by the different observations and models. This information would vanish when building differences between two data sources. For a more quantitative comparison the bi-monthly average profiles are shown in Figs. 11 and 12 where differences can easily be inferred by the reader. Moreover, there is no single data source covering the entire altitude range up to $10^{-3}$ hPa depicted in Figs. 9 and 10 what would be a necessary condition for a reference profile.

---

## Author Response (AR1)

**Reply to anonymous referee #1**

Review of doi: 10.5194/amt-2017-390

R. Rüfenacht et al. 2017

- blue: reviewer comments
- green: author replies

**General comments:**

- This manuscript presents inter-comparisons between a radiometer capable of horizontal wind measurements in the middle atmosphere (WIRA) and a lidar (ALOMAR RMR) capable of wind measurements in addition to temperature and aerosol properties. Intercomparisons are also made between these measurements and the ALOMAR meteor radar (although these measurements do not cover the same height region), as well as various models/re-analysis data sets. It is a useful study highlighting the capabilities of both the WIRA and the ALOMAR RMR to make useful measurements of wind in the upper stratosphere and lower mesosphere, a particularly difficult region of the atmosphere to measure wind in.

- In its current form the manuscript reads as three separate studies with a common linkage, the WIRA instrument, which is compared to the lidar and radar measurements and to various re-analysis data sets. Hence, I suggested the title change and in general suggest the Abstract, Introduction, and Conclusion focus on the validation of WIRA as the "hub" of the study, as for instance, there are not enough lidar measurements to make this a validation between WIRA and the lidar and poor/no overlap with the meteor radar.

It seems that the amount of lidar observations was not well appreciated here and we suspect that the manuscript did not make it clear enough. Therefore we extended our introduction near page 2, line 17 by "During this period, 518 hours of coincident measurements of sufficient duration[3] and an uninterrupted series of 187 hours of continuous day and night observations have been recorded." with the following footnote "[3]Only measurements longer than 5 hours are considered in this study in order to mitigate effects of the different pointing of the instruments (see Sect. 4) and to guarantee stable radiometer retrievals." Moreover we added the following information to the conclusions near page 13, line 7: "This part of the study is based on 518 hours of coincident observations by the ALOMAR RMR lidar and the microwave radiometer WIRA with individual recordings having a minimal duration of 5 hours."

We would like to point out that each instrument and model contributes with a valuable piece of information which has some advantage which cannot be achieved by the other contribution: For instance, it is clear that no middleatmospheric lidar, especially not at locations with frequent cloud cover can be used for continuous monitoring. Its main advantage is the high temporal and vertical resolution. On the other hand wind radiometry cannot offer the high resolution, e.g. to directly observe gravity waves, but thanks to its adverse-weather capabilities can achieve long term continuous monitoring. We therefore refuse to focus on one sole instrument as there is currently no "perfect" tool for assessing middle atmospheric wind. A multi-instrument approach is clearly beneficial. We also included models and meteor radar data to this study in order to get the most encompassing picture of the atmosphere possible. Only thanks to this variety of independent sources it is possible to gain some hints on which source appears to be biased from the "reality" at some time.

**Specific comments:**

• 1: Title. Currently: "Validation of middle-atmospheric wind in observations and models" does not read well, nor adequately describe the study. My suggestion: Validation of Microwave Radiometer Wind Measurements Using Active Remote Sensing and Models
Thank you for you suggestion. We feel you are uncomfortable with the term "validation" Therefore we changed the title to "Intercomparison of middle-atmospheric wind in observations and models" and the shorttitle to "Intercomparison of middle-atmospheric wind.' Your proposed title would suggest that wind radiometry is the only unvalidated data source while the others are all well established standards. This does however not correspond to reality as is exposed in the introduction of our manuscript. Therefore we think that our (modified) more encompassing title is justified and adequately descriebes the content of the manuscript.

• 2: I am not an expert in microwave techniques. That said I am uncomfortable with a new correction to the wind retrieval involving being introduced (a correction for mesospheric ozone) via a short description in an Appendix. The authors should write a more detailed manuscript on this improvement to the technique, including demonstrating the affect of various realistic mesospheric ozone profiles on their original and revised technique, as well as validation of the improvement by comparisons with measurements if possible. It is OK to describe the ozone correction in the text and apply it to the measurements, I don't think this paper should be a justification/validation of this correction since you are not specifically picking out examples and showing improvements in the results. Comparisons with/without this correction are important, but I see them as beyond the scope of this paper but requiring a more careful assessment/validation.
This improvement of the retrieval algorithm originates from the consideration of the secondary ozone maximum which had been previously neglected. The secondary ozone maximum is a known feature in the mesopause region and is described in many papers (e.g. Evans and Llewellyn, 1972; Hays and Roble, 1973; Smith and Marsh, 2005; Smith et al., 2008; Tweedy et al., 2013). Hence the improvement is due to a more accurate description of the physics in the radiative transfer model and not the result of some arbitrary tuning and there is no reason which would justify to revert back to a less accurate model. We have published another paper entirely devoted to this topic (Rüfenacht and Kämpfer,

2017) to which we refer in the retrieval description in the present manuscript by using a citation. It contains a detailed description on the underlying physics and the reason for the choice of the currently used a priori statistics. The paper also quantifies the effect of the modification to the retrieval setup using Monte Carlo simulations in its Figs. 4 to 8. Finally the difference between the wind retrievals with the old and the new setup based on measured data is shown in time series and diurnal cycle in comparison with ECMWF and SD-WACCM model data in Figs. 12 to 14.

The appendix to the present paper is not intended for the description of the retrieval but rather to illustrate the potential for future mesopause region wind measurements with a ground-based microwave radiometer. For more information on the improved retrieval algorithm the reader should refer to Rüfenacht and Kämpfer (2017). With our old formulation in the manuscript the reader may not have understood that the modifications to the retrieval suggested in Rüfenacht and Kämpfer (2017) are based on established knowledge in atmospheric physics and chemistry. Moreover tests of the quality of the upgraded retrieval in contrast to the legacy version through intercomparisons with models have already been presented in Rüfenacht and Kämpfer (2017). We tried to clarify these points in the manuscript near page 3 line 31: "Based on considerations to atmospheric physics/chemistry and radiative transfer as well as on the comparisons of the day/night differences in the radiometer observations and models presented in Rüfenacht and Kämpfer (2017) the authors judge now also the nighttime observations of mesospheric wind by WIRA to be largely bias-free. This quality is intended to be confirmed by the first instrumental intercomparisons carried out in the present study."

• 3: Section 2.1 Spatial averaging in WIRA could explain the poor meridional agreement in the mesosphere with the lidar. Changes in meridional flow, particularly at high latitudes, can be abrupt in latitude; the larger latitudinal spread of WIRA to determine a wind could be averaging across two very different flows, while the lidar is sampling the same one.

This was also one of our thoughts. However, we present a case study for the time of the largest differences from 4 to 6 February 2017 in the supplementary material in Fig. S6. We see this figure as a very strong indication that the different horizontal extent of the sampling volumes is not the reason for the dissent to the lidar although it has to be noted that Fig. S6 is based on ECMWF forecast data which may not resolve all small scale structures. This mark of caution was also added to the caption to Fig. S6: "Note that a global numerical weather prediction model as ECMWF is not expected to resolve all small scale structures and might smooth out some particularly strong spatial gradients. Nevertheless, in the light of the present figure, it appears higly unprobable that differeing horizontal sampling is responsible for the differences in mesospheric meridonal wind between lidar and radiometer in the period of 4 to 6 Feb 2017."

As seen from Fig. R1 most meteors are detected between 50 and 60° off-zenith hence the horizontal extent of the sampling volume is comparable to WIRA and much larger than for the lidar. Hence, to complete our discussion on vertical extnent of sampling volume we added to Sect. 4.1. near page 7 line 23: "The Andenes meteor radar obtains most meteor echoes from zenith angles between 50° and 60° leading to an average observation volume extent of about 160 km at an altitude of 70 km."

[Figure]

Figure R1: Histogram of the number of meteor detections per zenith angle by the Andenes meteor radar

We also extended our discussion concerning the possible reasons for the disagreements near page 9 line 25: "... could not be definitely identified. Although the radiometer and the meteor radar cover an observation volume of significantly larger extent than the lidar the discrepancy can most probably not be attributed to the different spatial sampling (see Supplement's Fig. S6). Nevertheless the substantial spread among the models and re-analyses indicate a rather heterogeneous atmosphere. A differing temporal evolution of the sensitivities of the lidar and the radiometer to these high altitudes might explain the dissent. Such effects could be introduced by temporally evolving cirrus or polar stratospheric clouds altering the transmission of the lidar signal or by variations of the mesospheric ozone concentration modulating the strength of the microwave emissions."

• 4: Section 3.4 Limitations of the geostrophic analysis should be discussed a bit further, pointing out the affect of curvature of the isobars and of baroclinic instability on the assumption of geostrophic balance.
We added the following lines to the manuscript in Sect. 3.4: "Note, that in this formulation friction, vertical advection and time tendency is neglected and that the geostrophic balance is assumed, i.e. the exact balance between Coriolis force and pressure gradient force. Therefore the geostrophic wind is directed parallel to the isobars and does not depend on curvature at all meaning that the air does not flow from high to low pressure. However, outside the tropics geostrophic wind can often be regarded as a reasonable approximation for the real wind."

• 5: End of Section 3: a semi-diurnal tide might average out but would a diurnal tide? What about tidal/mean wind/planetary wave interactions? Tidal structures can be cmoplicated at higher latitudes. Please give a more complete assessment of these affects in the manuscript.

[Figure]

Figure R2: Amplitude of the diurnal and semi-diurnal tide in middle-atmospheric winds as observed by the Andenes meteor radar.

The absence of local time precession (sun synchronous orbit) in the Aura orbit precludes the determination of tides in MLS data meaning that the ascending and descending node samples are stationary with respect to migrating tides in general. Thus the tidal impact should appear as constant offsets to the measurements at a particular location. It should also be noted that the amplitudes of all tidal components in the USLM are not comparable to what is generally observed by mesopause region radars. This can also be seen in Figs. R2 and R3. Last but not least tidal/mean wind/planetary wave interactions are included in the MLS data but so they are in the WIRA and meteor radar data in Figs. 9 to 12, which integrated over the whole day. Hence they are not expected to be responsible for large biases. We extended the discussion on the tidal impact on the MLS data in Sect 3.4: "Some marginal aliasing effects on MLS data from the migrating tides can not be excluded. However, since Aura is in a sun synchronous orbit, its samples are stationary with respect to migrating tides. These should appear as constant offsets to the measurements at a particular latitude. Especially the effect of the diurnal tide which appears to be the strongest tidal component in the middle atmosphere is strongly reduced by the averaging over the measurements during the satellite's overpasses in the ascending and descending orbit spaced by 12 hours. A more detailed discussion on the impact of tides on MLS measurements can be found for example in Lieberman et al. (2015) and Xu et al. (2009). It should also be remembered that, in contrast to the mesopause region, tides are usually weak in the stratosphere and lower mesosphere (e.g. Baumgarten et al., 2018; Kopp et al., 2015; Sakazaki et al., 2018)."

• 6: To continue on the previous point, the large differences between the meridional wind a greater heights could be due to Points 5 & 6. Please discuss this possibility.
We guess you mean Points 3 to 5. Thanks for your suggestions. For our replies to your suggestions and the modifications we made to the manuscript please refer directly to the previous points.

• 7: Section 4.2: you discuss random uncertainties, but what of the systematic uncertainties? Systematic uncertainties can have a large affect on a wind

[Figure]

Figure R3: Amplitude of the diurnal and semi-diurnal tide in middle-atmospheric winds as present in the MERRA re-analysis. Note that the diurnal tide is mostly averaged out in the geostrophic wind calculations by using MLS measurements from the ascending and descending orbits spaced by 12 hours.

measurement.

Systematic uncertainties for the radiometer have been assessed in Rüfenacht et al. (2014) and for the lidar in Baumgarten (2010) and Hildebrand et al. (2012) and in more detail in the dissertation by Hildebrand (2014). Nevertheless, we agree that unknown systematic errors can be a fundamental limitation to the qualification of the uncertainties of any physical measurement technique. This actually is the reason why cross-validations as presented in this paper are so important, because it is the only possibility to dissolve systematic errors of which nobody has thought of before. For such studies it is important that the different measurement techniques are as independent as possible from each other so that they do not share similar systematic errors. This is clearly the case for microwave radiometry, the lidar and the radar technique. Hence the small biases found in our cross-validations among the observations and in the comparisons to the models can be regarded as an evidence that none of the observation techniques suffers of large systematic errors.

• 8: Section 6 - more discussion of the poor agreement between WIRA and the meteor meridional radar wind is needed, the wind variations on either side of the "line" in Figures 9 and 10 are huge. Figure 13 attempts to argue this isn't so bad, but is not called out in the main body of the text, and no detailed explanation of what the "convolved version" of the measurements means.

As stated in the caption to Fig. 9 the trustworthy altitude range of the radiometer observations in Figs. 9 and 10 is between the grey lines varying with altitude over time (which has to do with differing observation conditions especially due to the tropospheric water content). Therefore WIRA and meteor radar observations on both sides of the 0.02 hPa line can only be compared to each other when the trustworthy altitude range or the wind radiometer reaches this altitude. In order to further clarify this point we added the following sentence to the caption of Fig. 9: "Radiometer data beyond this range are noticeably influenced by a priori assumptions should not be used for comparisons e.g. with meteor radar observations." Moreover we added this clarification to the meteor radar instrumen description near page 5 line 5:"...are adjacent to the uppermost levels

covered by WIRA and the RMR lidar, at least in good observation conditions (reasonably low tropospheric water content for the radiometer, no cirrus clouds for the lidar)." When this condition is satisfied, i.e. when the upper dark grey line does not lie lower than 0.02 hPa, we do not see any "huge" differences. This relatively good agreement (which is also mentioned in the conclusions as you request in your marked-up version of the document) can also be confirmed when looking at Figs. 11 and 12 which allow for a more quantitative comparison.

In Figs. 11 and 12, care shall be taken for the comparisons between meteor radar and the USLM measurements and model data in seasons with rapidly changing upper altitude limit for WIRA (see Figs. 9 and 10). In Figs. 11 and 12 this is the case for panels 1, 5 and 6. There the uppermost part of the USLM average profiles do not contain each day. Indeed USLM data at each altitude are only considered when the radiometer observations are judghed trustworthy at this level. In this way it is guaranteed that we only focus on simultaneous observations. On the ohter hand, meteor radar profiles are the average over all days because this approach is not possible for the meteor radar which samples another altitude range. Thus for time periods where the upper limit of the trustworthy altitude range is frequently below 0.02 hPa the averages for USLM and meteor do not rely on the same set of days. Moreover it must be noted that, in contrast to all USLM data, meteor radar winds are never convolved with WIRA's averaging kernels according to Eq. (2). As they do not cover the same altitude range a meaningful convolution is not possible.

These points have been clarified in the manuscript by adding the following statement to the caption of Fig. 11: "At the uppermost altitudes the raw, i.e. unconvolved, meteor radar wind profiles are shown. Due to the temporal variations of the upper altitude limit of the radiometer observations visible in Figs. 9 and 10 the sampling period of the meteor radar average wind can be rather different from the highest levels of middle-atmospheric data especially for the summer half-year, i.e. in the upper left, the lower centre and the lower right panel." We also added the following statement to the manuscript near page 11 line 2: "Similarly the measurement conditions influence the upper limit of WIRA's trustworthy altitude range. In Figs. 11 and 12 USLM data at each altitude are only considered when the radiometer observations are judghed trustworthy at this level. This guarantees that all USLM average profiles are based on simultaneous observations/data. As this approach is not possible for the non-overlapping altitude range of the meteor radar, its profiles are averages over all days. This may lead to slightly different temporal sampling between the USLM and the meteor radar data for the tree panels of the summer half-year when WIRA's uppermost trustworthy altitude is not constantly adjacent to the 0.02 hPa line (see Figs. 9 and 10). Moreover, it should be noted that in contrast to the USLM observations meteor radar winds are never convolved with WIRA's averaging kernels according to Eq. (2)." After clarification of all these points some re-organisation of this chapter was necessary in order to maintain the readability of the text. These modifications are visible in our marked-up version of the manuscript

Figure 13 has absolutely nothing to do with the other figures mentioned by you nor is it in the manuscript to argue anything about previous figures. Also a careful re-check of our manuscript did not show a possible source for such misunderstandings. Note also that the label of Fig. 13 will be changed to Fig. A1 (as it is actually part of the appendix) in the production.

To your comments concerning convolution: We agree that in the Appendix near page 14 line 26 "convolved version" is not precise enough and can confuse people who are not too familiar with microwave radiometry. Therefore we replaced it by the formulation "especially when these are convolved with WIRA's averaging kernels according to Eq. (2)." We judge that the explanations in the vicinity of Eq. (2) are sufficient for a good understanding, but the link to this formula was just not explicit enough in our formulation.

A related question was asked by you in the marked-up version of the manuscript which shall be answered here: The smoothing error is quantifying the uncertainty of the wind at a certain altitude due to the limited altitude resolution of the wind radiometer. It is generally calculated based on the assumption that the real wind follows the statistics of the a priori assumption and is thus inherently difficult to correctly estimate. The advantage of comparing wind radiometer profiles and convolved high-resolution profiles from lidar and models according to Eq. (2) is that the limited altitude resolution of the the radiometer is already considered (hence cannot affect the comparsion anymore) and the smoothing error with its uncertainties does not need to be estimated (see e.g. Rodgers, 2000).

• 9: Some modifications to the conclusion are suggested, and are included on the marked up copy of the manuscript.

We added our replies (in green) directly to your commented manuscript. It is provided as an author's reply supplement.

• 10: While I agree there is not an over-abundance of Rayleigh lidar wind measurements, there have been numerous papers by groups in France, the United States, and others since the 1980s (e.g. Chanin, Tepley, Meriwether, Keckhut). I believe there is also some comparisons between the lidars at La Reunion and the WIRA instrument? I suggest you mention some of this previous work in the Introduction.

We agree that this pioneering work is very important. It is also true that some more precision is needed in our manuscript. We therefore adapted the introduction. Near page 2 line 5 "techniques for wind observations in USLM" was replaced by "In the last decade two new techniques that achieve wind observations throughout the entire USLM and independently of daylight conditions became operational" Further we added the following sentence "While wind radiometry was developped from scratch, the lidar technique could benefit from earlier works on nighttime stratospheric wind measurements by lidar (Chanin et al., 1989; Souprayen et al., 1999; Tepley, 1994; Friedman et al., 1997). Due to the novelty of the two approaches..."

We are sorry for not having cited this literature in the in the initial submission because we were only focussing on lidars covering the entire altitude of interest, which we agree was not giving enough credit to the pioneers. Please note that the initial statement by Tepley et al. (1991) that nighttime wind measurements were possible throughout the entires USLM was revoked by Tepley (1994) because of saturation/memory effect problems with their photomultuplier tubes which resulted in spurious high-altitude signals. John Meriwether has published significant contributions of USLM temperature observations and their interpretation but he has, to our knowledge, not written a first-author paper on wind observations in the USLM. Meriwether and Gerrard (2004) rather pointed out that such observations are needed. Similarly, Philippe Keckhut was involved in

many of the french lidar work but we have not found a first author publication describing another instrument than the previously mentioned.

From personal contacts we know that there are currently only very few (night-time, and no daytime) wind retrievals from the new lidar on Maïdo on La Réunion longer than say 3 hours available for intercomparison with the co-located second generation wind radiometer WIRA-C. This is in clear contrast to the 518 hours of measurements longer than 5 hours which we present in our paper. It should moreover be noted that the La Réunion lidar only reaches up to 50 km in contrast to the study presented in our paper which also includes validation in the lower mesosphere. Data intercomparisons from La Réunion have never been published up to now. Based on your previous comments we understand that you agree that a high amount of measurement hours is necessary for meaningful intercomparisons. Therefore we can not (yet) cite any intercomparisons from La Réunion.

• 11: The review of the capabilities of radars in this region is incomplete and should be revised. Some VHF radars can measurement wind below 70 km, but this capability is latitudinally dependent. The Japanese Antarctic radar, Pansy, has made wind measurements down to 55 km. MF radars routinely get measurements down to 60 km. I would suggest a paragraph explaining what the radars can and can't do, and how WIRA and the lidar measurements complement the radar work. Please see the recent book by Hocking et al, which gives many of these references (Hocking, W., Röttger, J., Palmer, R., Sato, T., & Chilson, P. (2016). Atmospheric Radar: Application and Science of MST Radars in the Earth's Mesosphere, Stratosphere, Troposphere, and Weakly Ionized Regions. Cambridge: Cambridge University Press. doi:10.1017/9781316556115).

Thank you for the suggested literature. We agree that our review of radar techniques was not complete. Nevertheless it must be noted that none of the radar techniques including MF radars does routinely measure winds at 60 km altitude. It is true that during solar proton events and other strong particle precipitations observations down to 60 km have been reported. However such measurements are far from being routine observations, they are possible in less than 10% of the time. MF radars typically deliver measurements in a region around 70-90 km. A detailed comparison to meteor radar data can be found in Wilhelm et al. (2017) which is also cited in the manuscript.

The introduction has been extended to give a more comprehensive overview on radar capabilities near page 1, line 15: "The widely used radar techniques can usually not assess the USLM due to the lack of backscatterers (charged particles, turbulent structures at scales of the radar wavelength). Only in the event of strong particle precipitations have measurements down to 60 km been reported (e.g. Nicolls et al., 2010; Shibuya et al., 2017, for an encompassing overview on radar observation techniques refer to e.g. Hocking (2016))."

• Other minor suggestions and changes are indicated on the marked-up copy of the manuscript attached.

Thank you for all these suggestions. We added our replies (in green) directly to your commented manuscript. It is provided as an author's reply supplement.

- Page 2 section 2 No reference is performed on Radiosondes. Figure 1 suggest that no RS exist at Andenes. Is it true?

It is true that there are no routine radiosoundings at Andenes. Some soundings have been performed on campaign basis, but not during the time period under investigation. This does however not negatively impact the present study because radiosondes generally reach top altitudes of 30-35 km and would thus not provide significant altitude overlap with the lidar or the wind radiometer anyway. The major general circulation models broadly assimilate radiosoundings and are validated against observations at these altitudes so that this is not the focus of the present study.

- Figure 9 and 10 Wind differences are difficult to estimate from these figures. Is it possible to provide differences instead of wind profiles?

We completely agree that it is hard to draw quantitative comparisons from Figs. 9 and 10. The primary aim of these figures is to present the time series showing how the dynamical patterns and their temporal evolution are captured by the different observations and models. This information would vanish when building differences between two data sources. For a more quantitative comparison the bi-monthly average profiles are shown in Figs. 11 and 12 where differences can easily be inferred by the reader. Moreover, there is no single data source covering the entire altitude range up to $10^{-3}$ hPa depicted in Figs. 9 and 10 what would be a necessary condition for a reference profile.

[revised manuscript text omitted]